# MDF is a conserved splicing factor and modulates cell division and stress response in *Arabidopsis*

Cloe de Luxán-Hernández[1], Julia Lohmann[1], Eduardo Tranque[2], Jana Chumova[3], Pavla Binarova[3], Julio Salinas[2], Magdalena Weingartner[1]

The coordination of cell division with stress response is essential for maintaining genome stability in plant meristems. Proteins involved in pre-mRNA splicing are important for these processes in animal and human cells. Based on its homology to the splicing factor SART1, which is implicated in the control of cell division and genome stability in human cells, we analyzed if MDF has similar functions in plants. We found that MDF associates with U4/U6.U5 tri-snRNP proteins and is essential for correct splicing of 2,037 transcripts. Loss of MDF function leads to cell division defects and cell death in meristems and was associated with up-regulation of stress-induced genes and down-regulation of mitotic regulators. In addition, the *mdf-1* mutant is hypersensitive to DNA damage treatment supporting its role in coordinating stress response with cell division. Our analysis of a dephosphomutant of MDF suggested how its protein activity might be controlled. Our work uncovers the conserved function of a plant splicing factor and provides novel insight into the interplay of pre-mRNA processing and genome stability in plants.

## Introduction

For plants as sessile organisms, the only way to survive in an ever-changing environment is to continuously adjust cell division and growth to environmental signals and stress conditions. This is especially important in meristematic cells, which are responsible for the continuous and error-free production of new cells for all plant organs. As in all other multicellular organisms, a conserved cell cycle machinery, which is controlled by the activities of conserved cyclin-dependent kinases and cyclin complexes, drives plant cells through the cell cycle. They ensure the controlled progression through the individual phases of the cell cycle, namely, gap phase 1 (G1), DNA replication (S), gap phase 2 (G2), and mitosis (M) (Shimotohno et al, 2021).

To maintain genome integrity during cell division, cells need to respond to DNA damage signals by activating the so-called DNA damage response (DDR) pathway. The DDR is controlled by two conserved kinases, Ataxia-telangiectasia mutated (ATM) and ATM and RAD3-related (ATR). Upon activation, they phosphorylate a plant-specific NAM-ATAF1/2-CUC2 (NAC) transcription factor, SUPPRESSOR OF GAMMA RESPONSE 1 (SOG1). Once being activated, SOG1 induces factors that lead to inhibition of cell cycle progression, activation of DNA repair (Culligan et al, 2006; Yoshiyama et al, 2009; Yi et al, 2014), and induction of cell death, if the cellular damage cannot be repaired (Furukawa et al, 2010; Johnson et al, 2018). SOG1-mediated cell cycle arrest was shown to occur at the G2 phase of the cell cycle through transcriptional induction of cyclin-dependent kinase inhibitors (Yi et al, 2014; Ogita et al, 2018) and the NAC transcription factors ANAC044 and ANAC085. This leads to stabilization of MYB3R repressors (Rep-MYBs) which inhibit the expression of genes required for the transition to mitosis (Takahashi et al, 2019).

Regulation of precursor-messenger RNA (pre-mRNA) splicing is a very effective mechanism that allows cells to rapidly adjust their transcriptional program to stressful conditions. In all eukaryotic cells, pre-mRNA splicing is catalyzed by the spliceosome, which is formed by the ordered interaction of four small ribonucleoprotein particles (snRNPs), named U1, U2, U4/U6, and U5 snRNPs, and additional splicing factors with the pre-mRNA (Will & Luhrmann, 2011). The splicing process is triggered by binding of U1 and U2 snRNPs to the 5'splice site and branching point, respectively. Subsequently, pre-assembled U4/U6 snRNPs interact with U5 snRNPs to form the U4/U6.U5 tri-snRNP. Only under splicing conditions, the tri-snRNP becomes integrated into the spliceosome forming the pre-catalytic splicing complex. Major structural changes during which numerous U4/U6– and U5 snRNP–associated proteins including the LSM2-8 complex are released lead to formation of the catalytically active spliceosome. After completion of the splicing reactions, the spliceosome is dismantled and the tri-snRNP is re-formed and takes part in new rounds of splicing (Wan et al, 2020; Wilkinson et al, 2020).

The importance of splicing proteins during DDR in animal and human cells is well established, and a large part of the transcriptional reprogramming during DDR was shown to be mediated by mechanisms regulating mRNA processing and transcript stability

[1]Institute of Plant Sciences and Microbiology, University of Hamburg, Hamburg, Germany    [2]Departamento de Biotecnología Microbiana y de Plantas, Centro de Investigaciones Biológicas "Margarita Salas" (CSIC), Madrid, Spain    [3]Institute of Microbiology of the Czech Academy of Sciences, Prague, Czech Republic

Correspondence: Magdalena.weingartner@uni-hamburg.de

(Dutertre et al, 2011; Boucas et al, 2012). In plants, the role of splicing factors during response and adaption to various abiotic stress conditions is well established (Laloum et al, 2018; Ling et al, 2021; Martin et al, 2021). For instance, for several core components of the spliceosome, such as the LSM2-8 complex or the U5-snRNP protein STABILIZED1 (STA1), a crucial role during adaption to salt and temperature stress was established (Carrasco-Lopez et al, 2017; Kim et al, 2017). In addition, serine/arginine (SR)–rich SR proteins were shown to act as important splicing regulators during environmental responses. One prominent example is the SR-like protein SR45 that is an important regulator of developmental processes (Ali et al, 2007; Carvalho et al, 2010; Chen et al, 2019) and stress tolerance (Carvalho et al, 2016; Albaqami et al, 2019). However, it is not well understood to what extent plants use the splicing machinery for regulating DDR pathways (Nimeth et al, 2020).

The purified tri-snRNP from human cells contains about 30 proteins, and corresponding *Arabidopsis* homologs have been annotated (Koncz et al, 2012). One of them is MERISTEM DEFECTIVE (MDF), which is the *Arabidopsis* homolog to the human protein SQUAMOUS CELL CARCINOMA ANTIGEN RECOGNIZED BY T CELLS 1 (SART1). SART1 is only found in assembled tri-snRNP complexes and released form the spliceosome before its catalytic activation (Hacker et al, 2008). It was shown to be necessary for the association of the tri-snRNP with the pre-spliceosome but not for the stability of the tri-snRNP itself (Makarova et al, 2001). Likewise, its yeast homolog SNU66 is copurified with U4/U6.U5 tri-snRNPs (Gottschalk et al, 1999; Stevens & Abelson, 1999) and is important for efficient pre-mRNA splicing in *Saccharomyces cerevisiae* (van Nues & Beggs, 2001). *SART1* was originally identified as a tumor antigen in a range of cancers recognized by T cells (Kikuchi et al, 1999). Its expression was highly induced in different cancer cell lines and cancer tissue (Allen et al, 2012). Moreover, siRNA-based silencing revealed SART1 as an essential factor for cell division whose depletion was associated with cell division defects and induction of apoptosis (Kittler et al, 2004). MDF shares 38.9% overall sequence similarity with SART1. Based on its N-terminal putative RS domain–containing arginine residues alternating with serine, glutamate, or aspartate dipeptides, MDF is like SART1, an SR-like protein (Neugebauer et al, 1995; Blencowe et al, 1999; Casson et al, 2009). Published data have shown that MDF expression occurs mainly in dividing cells and that the *mdf-1* mutant showed impaired primary root development. These defects were associated with reduced cell division and cell elongation and mis-expression of genes involved in auxin regulation (Casson et al, 2009). Two independent phospho-proteomic screens have shown that MDF was phosphorylated at Serine 22 (S22) in proliferating cells (Roitinger et al, 2015; Waterworth et al, 2019), and this phosphorylation was increased upon DNA damage treatment (Roitinger et al, 2015).

In this study, we analyzed if MDF has a conserved function in pre-mRNA splicing in *Arabidopsis* and how this was related to its role during cell division and meristem development. We show that MDF is associated with plant tri-snRNP complexes, localizes to nuclear speckles, and is important for correct splicing of numerous genes involved in developmental and signaling processes. MDF loss of function led to cell cycle arrest at the G2/M transition and increased endoreduplication levels. This phenotype was associated with altered expression of proliferation-associated genes,

down-regulation of mitotic genes, and constitutive up-regulation of stress-related genes. Our analysis of a dephosphomutant version of MDF showed that MDF phosphorylation at S22 is important for its function in maintaining cell division activity in meristems. The interplay between pre-mRNA splicing and plant DDR was further addressed by assessing the sensitivity of mutants with impaired splicing activity to DNA damaging conditions. Thus, our work provides new insight into the role pre-mRNA processing in controlling cell division, development, and the expression of stress response genes in plants.

# Results

### The short root phenotype of *mdf-1* and *mdf-2* is associated with cell cycle arrest at the G2/M transition

To assess how MDF loss of function affects cell proliferation in plants, we analyzed root apical meristem (RAM) development in WT, the two previously described T-DNA insertion lines for MDF *mdf-1* and *mdf-2* (Casson et al, 2009) and a complementation line for *mdf-1* harboring a construct in which the genomic region of *MDF* including 3,000 bp upstream of the start ATG was fused at the C-terminus to GFP (*mdf-1::pMDFMDFg*). Root length measurement over a time course from 1 to 10 days after germination (dag) confirmed that in both mutants, root growth was strongly delayed compared with WT and *mdf-1::pMDFMDFg* (Fig 1A). By confocal imaging of propidium iodide (PI)–stained root tips (Fig 1B), we verified that in both *mdf* mutant lines, the cell division zone of the RAM was reduced as compared with WT and the complementation line. Quantification of the number of dividing cells in the cortical cell layer proved that the shorter meristem was because of a reduced number of dividing cells (Fig 1C). Furthermore, the PI staining revealed that the absence of MDF lead to the accumulation of dead cells in the cell division zone of the RAM, which was not observed in the WT or the complementation line (Fig 1B and D). The presence of dead cells in the meristem of *mdf-1* but not in WT or the complementation line was further confirmed by confocal analysis of root tips stained with fluorescein diacetate (FDA) (Fig S1A). Spontaneous induction of cell death is indicative for impaired genome stability and can be caused by defects during progression through mitosis (Hu et al, 2016; Nisa et al, 2019). To understand at which stage of cell division meristematic cells were arrested, we performed immunolabelling of microtubules in root tip cells. The quantification of microtubule arrays of six dag seedlings revealed that the number of cells showing mitotic microtubular arrays was reduced in *mdf-1* and *mdf-2* compared with WT. In contrast, the number of cells exhibiting a preprophase band marking the G2/M transition was significantly increased (Figs 1E and S1B), indicating that the transition from G2 phase of the cell cycle to mitosis was impaired in the mutants. Genome instability and decreased division rates were shown to be associated with the premature onset of endoreplication (Adachi et al, 2011). Therefore, we measured the nuclear DNA content in 10 dag seedlings of WT, *mdf-1*, and *mdf-2* by flow cytometry and found that in both mutant lines, the level of endoreplication was significantly increased (Figs 1F and S1C). We also assessed shoot apical meristem activity by quantifying the number of seedlings that formed true

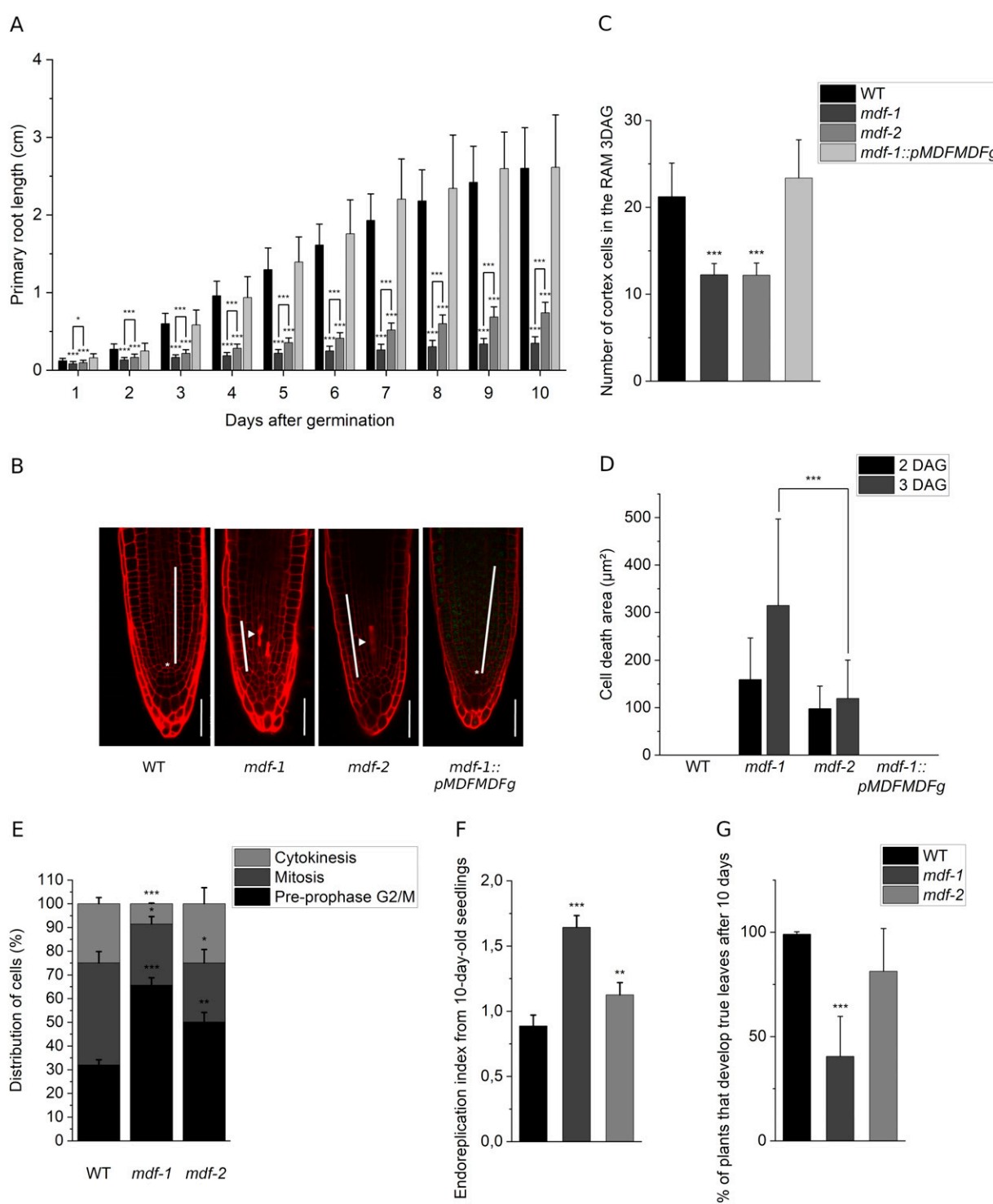

**Figure 1. Short root phenotype of *mdf-1* and *mdf-2* is associated with cell cycle arrest at the G2/M transition.**
**(A)** Primary root length from 1 until 10 days after germination (dag) of WT (n = 62, 137, 97, 74, 74, 75, 75, 74, and 75), *mdf-1* (n = 48, 65, 69, 55, 53, 41, 50, 44, and 45), *mdf-2* (n = 46, 57, 63, 51, 51, 51, 50, 52, 49, and 50), and *mdf-1::pMDFMDFg* (n = 72, 160, 161, 166, 223, 68, 66, 63, and 148). Statistical significance was determined in comparison to WT and *mdf-1*. **(B)** Representative confocal images of propidium iodide-stained root tips of WT, *mdf-1*, *mdf-2*, and *mdf-1::pMDFMDFg* 3 dag seedlings. Asterisk shows the quiescent center (QC). Arrowhead marks dead cells. White bar indicates the division zone. Scale bar: 50 μm. **(C)** Number of dividing cells in the cortical layer of three dag seedlings of WT (n = 40), *mdf-1* (n = 36), *mdf-2* (n = 21), and *mdf-1::pMDFMDFg* (n = 24). Statistical significance was determined in comparison to WT. **(D)** Quantification of the cell death area of root tips of WT (n = 21 and n = 22), *mdf-1* (n = 21 and n = 95), *mdf-2* (n = 33 and n = 59), and *mdf-1::pMDFMDFg* (n = 19 and n = 25) lines at 2 and 3 dag. **(E)** Analyses of immunolabeled mitotic microtubular arrays in root tips of WT (n = 348), *mdf-1* (n = 269), and *mdf-2* (n = 284) six dag seedlings. Percentual distribution of cells accumulated in G2/M with preprophase bands (G2/M, pre-prophase); pro, meta, and anaphase spindle (mitosis); and with phragmoplast (cytokinesis) was determined.

leaves at 10 dag. In this assay, only *mdf-1* but not *mdf-2* showed a significant difference to WT (Fig 1G), indicating that in *mdf-1*, not only root but also shoot meristem development was affected. As published previously, the insertion in *mdf-1* is located in intron 9, whereas in *mdf-2*, it lies within exon 9 (Casson et al, 2009). By RT-qPCR analyses, we found that in both mutant lines, the region upstream of the T-DNA insertion sites was still expressed, whereas the part of the gene downstream of the insertion sites, which includes the conserved SART1 domain, was not expressed in *mdf-1* but still highly accumulated in *mdf-2* (Fig S1D and E). This truncated version of MDF might still be partially functional which would explain the phenotypic difference between *mdf-1* and *mdf-2* mutant lines (Figs 1A–G and S1C). In summary, our phenotypic analyses showed that the defective meristem development in *mdf* mutants was associated with reduced cell division rates because of a G2-specific cell cycle arrest, genome instability, and increased levels of endoreplication.

## MDF associates with U4/U6.U5 tri-snRNP proteins

To assess if MDF as its human homolog hSART1 associates with the U4/U6.U5 tri-snRNP complexes in plant cells, we analyzed its interaction with LSM8, a core component of the tri-snRNP. LSM8 defines and confers specificity to the nuclear heteroheptameric ring complex LSM2-8 that binds and stabilizes the U6 small nuclear RNA, which is part of the U6 snRNP and, therefore, of the U4/U6.U5 tri-snRNP complex (Beggs, 2005; Tharun, 2009). To show that MDF was associated with this complex, we performed co-immunoprecipitation experiments, in which LSM8-GFP and putative interaction partners were co-immunoprecipitated using an anti-GFP antibody followed by tandem mass spectrometry (IP-MS/MS). MDF was among the proteins that specifically and abundantly co-purified with LSM8-GFP in three independent experiments (Table 1). To confirm the interaction, we performed bimolecular fluorescence complementation (BiFC) experiments using constructs harboring full-length ORF of *MDF* and *LSM8* fused at the N-terminus to either the C-terminal or N-terminal half of YFP. As expected, co-expression of these constructs produced a bright YFP-derived fluorescence signal, which was mainly seen in the nucleus (Fig 2A). No fluorescence was detected in control experiments using MDF and LSM1a ORFs (Fig 2A). Next, we used fluorescence resonance energy transfer (FRET)–based fluorescence lifetime imaging to test for colocalization and physical interaction of MDF-GFP with LSM8-mCherry in tobacco leaf epidermis cells. We found that MDF-GFP and LSM8-mCherry signals overlapped in the nucleoplasm, whereas only MDF-GFP was concentrated in the nucleolus and in nuclear condensates (Fig 2B). For FRET-FLIM measurements, constructs fused to GFP were used as donors and plasmids containing the fluorescent mCherry tag acted as acceptors. Our measurements revealed that there was no significant difference in the fluorescence lifetime of MDF-GFP in cells expressing MDF-GFP alone

compared with those expressing both MDF-GFP and LSM8-mCherry. However, consistent with previous results (Perea-Resa et al, 2012), a significant change in fluorescence lifetime occurred upon co-expression of LSM8-GFP and LSM2-mCherry (Fig 2C). These data indicated that LSM8 physically interacted with LSM2, as expected, but not with MDF. In addition, yeast 2-hybrid (Y2H) experiments were carried out and confirmed the physical interaction between LSM8 and LSM2 but not between LSM8 and MDF (Fig S2A). Human SART1 was shown to physically interact with the tri-snRNP component hPERP6 (Liu et al, 2006). Therefore, we also tested if MDF interacts with the plant homolog of hPERP6 named STA1 (STABILIZED1) and found that these two proteins physically interact in yeast (Figs 2D and S2B). We next tested whether the nuclear condensates, which were seen in MDF-GFP–expressing cells, represented nuclear speckles containing spliceosome proteins. To this end, MDF-GFP was co-expressed with SR45-mCherry, which was previously shown to localize, like many other splicing proteins, to nuclear speckles in plant cells (Ali et al, 2008). Indeed, we found that 76% of cells co-expressing MDF-GFP and SR45-mCherry showed nuclear condensates (n = 40). In each of these cells, the signals for MDF-GFP and SR45-mCherry overlapped (Fig 2E). Together, these data showed that MDF was a nuclear protein that associates with plant tri-snRNP complexes and colocalizes with other spliceosome proteins in nuclear speckles.

## MDF is important for correct splicing of transcripts involved in transcriptional control and signaling

To investigate if MDF loss of function was associated with defects in pre-mRNA splicing, we performed a high-coverage RNA-seq analysis on 12-d-old WT and *mdf-1* seedlings in three biological replicates. ~50 million 150–base pair (bp) paired-end reads per sample were generated in a NovaSeq 6000 platform. 2,945 differential splicing events corresponding to 2,037 genes were identified in the *mdf-1* background (Tables S1–S5). Among these altered splicing events, almost two-thirds were in the category intron retention (IR: ≈ 64%), whereas the categories exon skipping, alternative 5′ splice site, alternative 3′ splice site, and mutually exclusive exon were much less abundant (Figs 3A and S3). We performed a Gene Ontology (GO) term enrichment analysis based on the biological process ontology for genes showing increased IR defects in *mdf-1* (Fig 3B and Table S6). Genes associated with the GO terms "transcription," "protein phosphorylation," "protein modification," and "regulation of gene expression" were overrepresented. This indicated that MDF had an important function in controlling the correct splicing of transcripts involved in cellular signaling processes and transcriptional control. To gain more insight into the role of MDF in splicing regulation, we compared the altered splicing events identified in *mdf-1* with available RNA-seq data from mutants for three other components of the plant tri-snRNP in which a similar material (light grown seedlings) was used. These were *lsm8-1*

---

**(F)** Endoreplication index determined from flow cytometry data measured in WT (n = 7), *mdf-1* (n = 3), and *mdf-2* (n = 4) 10 dag seedlings. **(G)** Percentage of plants developing true leaves after 10 d for WT, *mdf-1*, and *mdf-2*. Average of at least four independent experiments with at least 20 plants per line, condition, and experiment is represented. Average ± SD is represented. *$P < 0.05$; **$P < 0.005$; ***$P < 0.0005$ as determined by a two-tailed *t* test.

**Table 1.   Mass spectrometry results of MDF protein co-immunoprecipitating with LSM8-GFP.**

| Accession | Description | Score | Coverage |
|---|---|---|---|
| R1 | | | |
| Q9LFE0 | SART-1 family protein DOT2 OS = *Arabidopsis thaliana* GN = DOT2 PE = 1 SV = 1 - [DOT2_ARATH] | 162,4486099 | 6,59 |
| | A2 | Sequence | Protein Group Accessions |
| | High | GLNEGGDNVDAASSGK | Q9LFE0 |
| | High | IQGQTTHTFEDLNSSAK | Q9LFE0 |
| | High | NSDTPSQSVQR | Q9LFE0 |
| | High | EASALDLQNR | Q9LFE0 |
| R2 | | | |
| Q9LFE0 | SART-1 family protein DOT2 OS = *Arabidopsis thaliana* GN = DOT2 PE = 1 SV = 1 - [DOT2_ARATH] | 198,5233817 | 7,93 |
| | A2 | Sequence | Protein Group Accessions |
| | High | GLNEGGDNVDAASSGK | Q9LFE0 |
| | High | IFEEQDNLNQGENEDGEDGEHLSGVK | Q9LFE0 |
| | High | NSDTPSQSVQR | Q9LFE0 |
| | High | VVEGGAVILTLK | Q9LFE0 |
| R3 | | | |
| Q9LFE0 | SART-1 family protein DOT2 OS = *Arabidopsis thaliana* GN = DOT2 PE = 1 SV = 1 - [DOT2_ARATH] | 210,6966728 | 12,2 |
| | A2 | Sequence | Protein Group Accessions |
| | High | GLNEGGDNVDAASSGK | Q9LFE0 |
| | High | IFEEQDNLNQGENEDGEDGEHLSGVK | Q9LFE0 |
| | High | NSDTPSQSVQR | Q9LFE0 |
| | High | mLPQYDEAATDEGIFLDAK | Q9LFE0 |
| | High | VVEGGAVILTLK | Q9LFE0 |
| | High | KPESEDVFmEEDVAPK | Q9LFE0 |

Results of the MS analyses showing the score, coverage, and sequence of the peptides that were identified for MDF in each of the three Co-IP experiments using LSM8-GFP as bait.

(Carrasco-Lopez et al, 2017), *rdm16-4* (Lv et al, 2021), and *brr2a-2* (Mahrez et al, 2016). The *rdm16-4* mutant has a point mutation in RDM16, encoding pre-mRNA splicing factor 3 (PRP3), which is part of the U4/U6 snRNP (Koncz et al, 2012). *BRR2a* encodes a RNA helicase and is not only part of the tri-snRNP but also remains associated with the catalytically active spliceosome (Bessonov et al, 2008). In each of these mutants, the most abundantly detected altered splicing event was IR. No representative overlap was found among the genes showing increased IR in all three mutants (Fig 3C). These data indicated that different splicing factors seem to have distinctive functions in splicing regulation. Although approximately half of the detected splicing defects were specific to *mdf-1* with enrichment in the GO terms "nucleobase-containing compound metabolic process" and "RNA metabolic process" (Fig 3B), specific GO categories relevant for its putative role in cellular signaling and transcriptional control were shared with other splicing mutants (Fig 3B). Those were "regulation of biological process" with *lsm8-1* (Table S7), "protein phosphorylation" with *rdm16-4* (Table S8), and

"RNA biosynthetic process" with *brr2a-2* (Table S9). Thus, each of these three splicing factors was required for splicing of those overlapping targets, whereas correct splicing of many targets affecting RNA and DNA metabolism seemed to specifically require the function of MDF.

To validate the RNA-seq data, we analyzed IR of selected targets by RT-qPCR on RNA samples isolated from 12-d-old seedlings of WT *mdf-1*, *mdf-2* and the complemented line (*mdf-1::pMDFMDFg*). Of the analyzed genes, five were involved in cell cycle regulation (*CyclinP3-2*, *CyclinB2-2*, *KRP6*, *KRP2*, and *SKP2B*) (Fig 3D), three had a role in splicing (*SR30*, *SR45*, and *RSZ33*) (Fig 3E), five were known as transcriptional regulators (*PFA4*, *MYB4R1*, *MYB3R3*, *MYBL*, and *REM30*) (Fig 3F), and one was associated to DNA repair processes (*RAD51D*) (Fig 3G). Among the 14 tested genes, we found for eight genes a significant increase in intron retention in RNA samples for both *mdf-1* and *mdf-2* as compared with WT. For six genes, a significant difference could only be confirmed in *mdf-1* but not in *mdf-2*. Intron retention was reversed in 13 of the targets in the

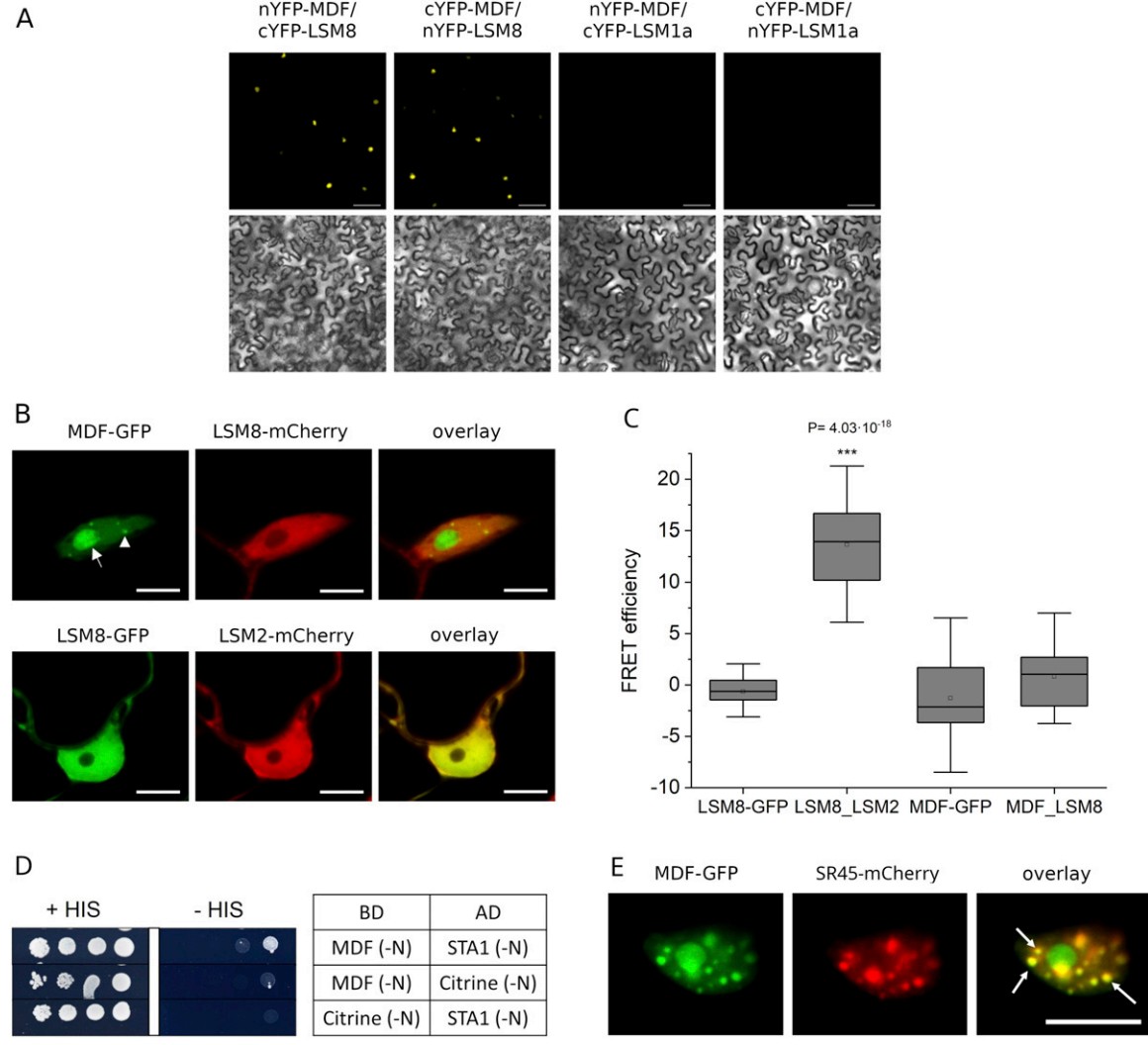

**Figure 2. MDF interacts with LSM8 and localizes to nuclear speckles.**
**(A)** Representative confocal images of BiFC assays showing reconstitution of YFP fluorescence in the nucleus of epidermal cells of *Nicotiana benthamiana* leaves in which nYFP-MDF and cYFP-LSM8 as well as cYFP-MDF and nYFP-LSM8 are co-expressed. No fluorescence is seen upon co-expression of nYFP-MDF and cYFP-LSM1a or cYFP-MDF and nYFP-LSM1a. Upper panels show fluorescence images and lower panels show bright field images. Scale bar: 75 μm. **(B)** Representative confocal images showing GFP (green) and mCherry (red)-derived fluorescence upon co-expression of MDF-GFP and LSM8-mCherry in *Nicotiana benthamiana* leaf epidermis cells. MDF localizes to the nucleolus (arrow) and co-localizes with LSM8-mCherry in the nucleoplasm. Scale bar: 9.7 μm. **(C)** Box graphs representing FLIM-fluorescence resonance energy transfer efficiencies (%) measured in *Nicotiana benthamiana* leaf epidermis cells expressing LSM8-GFP alone or in combination with LSM2-mCherry or MDF-GFP alone or together with LSM8-mCherry. Fluorescence resonance energy transfer efficiency was significantly higher in samples co-expressing LSM8-GFP and LSM2-mCherry (*$P$ < 0.0001; two-tailed $t$ test). **(D)** Yeast two-hybrid (Y2H) assay showing physical interaction between MDF and STA1. Growth of serial dilutions of yeast colonies was followed on medium without tryptophan and leucine (+HIS) and selective medium without tryptophan, leucine, and histidine (−HIS). BD, DNA-binding domain; AD, activation domain; interactions with BD and AD vectors containing citrine were used as negative controls. **(E)** Representative confocal images showing GFP (green) and mCherry (red)-derived fluorescence upon co-expression of MDF-GFP and SR45-mCherry in *Nicotiana benthamiana* leaf *epidermis* cells. Arrows indicate nuclear speckles in which both fluorescence signals overlap.

complemented line, further verifying that the splicing defects observed in these genes are very likely caused by the absence of MDF. These data showed that like its human and yeast homologs, MDF functions as a pre-mRNA splicing factor in plants.

### Loss of MDF function leads to altered expression of a large number of genes involved in stress response and cell cycle control

To understand how loss of MDF function affected gene expression, we used our RNA-seq data to detect genome-wide transcriptional changes. We found that the total number of genes that showed significantly altered transcript levels in *mdf-1* compared with WT (without setting any threshold for fold change in transcript levels) encompassed 7,516 genes that were up-regulated and 7,872 genes that were down-regulated (Fig 4A and Table S10). A pathway enrichment analysis among the significantly up-regulated genes revealed an enrichment of genes associated with stress responses. Among the top enriched GO categories were "abiotic and biotic stimuli," "temperature, salt and oxidative stress," and "defense response" (Fig 4B and Table S11). Interestingly, among the

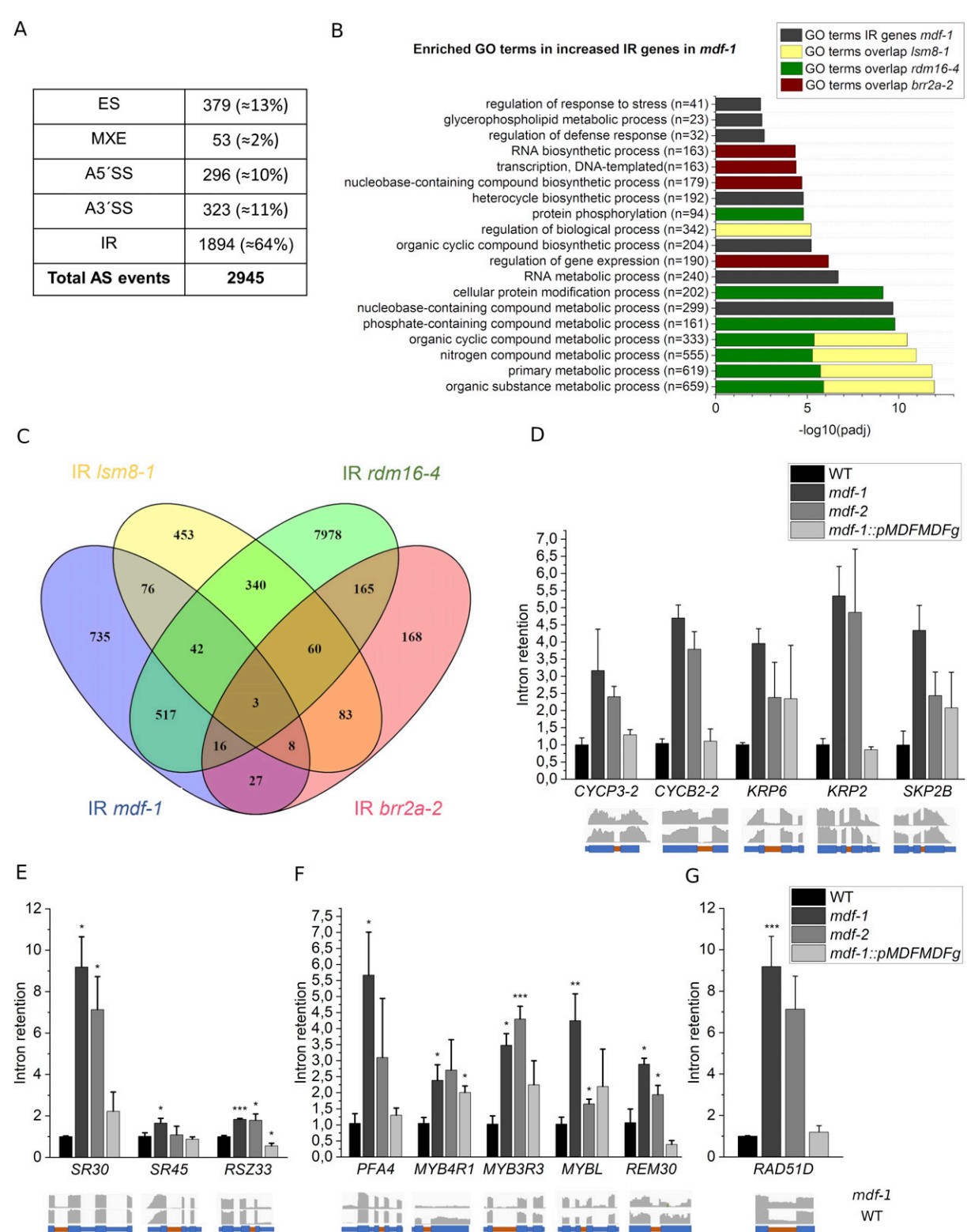

**Figure 3. MDF is important for correct splicing of transcripts involved in transcriptional control and signaling.**
**(A)** Quantification of alternative splicing (AS) events (ES, exon skipping; MXE, mutually exclusive exon; A5'SS, alternative 5' splice site; A'3SS, alternative 3' splice site; IR, intron retention) identified in *mdf-1* in respect to WT. **(B)** 20 most representative top significant biological process GO terms enriched in intron retained targets identified in *mdf-1*. Coloring indicates enriched biological processes found in targets showing IR in *mdf-1* and *lsm8-1* (yellow), *mdf-1* and *rdm16-4* (green), and *mdf-1* and *brr2a-2* (red). **(C)** Venn diagram representing the overlap between significantly increased IR targets between *mdf-1*, *lsm8-1*, *rdm16-4*, and *brr2a-2*. **(D, E, F, G)** RT-qPCR analysis to confirm increased IR events found by RNA-seq in seedlings of WT, *mdf-1*, *mdf-2*, and *mdf-1::pMDFMDFg* on genes involved in cell cycle (D), pre-mRNA splicing (E),

down-regulated genes, we found a significant enrichment of pathways involved in cell division and developmental processes. These top GO categories for down-regulated genes included "cell cycle," "cell division," and "chromosome organization," as well as "cell differentiation," "meristem development," and "root and shoot system development" (Fig 4C and Table S12). In addition, by comparing the total number of DEG with the total number of genes showing IR, we found that there was a statistically significant overlap of 907 genes (64% of all genes showing IR) (Fig 4D). Of these, more than 50%, namely, 501 genes were down-regulated (Fig 4D). According to our statistical analysis, the overlap between mis-spliced genes and down-regulated genes was significant, whereas no significant overlap was found for mis-spliced genes and up-regulated genes (Fig 4D). This observation correlated with the finding that IR might lead to introduction of a premature termination codon and subsequent targeting of transcripts to the nonsense mediated decay machinery, thus reducing their abundance (Filichkin et al, 2010). Interestingly, one of the biological processes enriched among the transcripts that were down-regulated and showed IR was "regulation of gene expression." Together, these data showed that loss of MDF function has a major impact on gene expression and that it is important for correct expression of genes involved in cell proliferation and developmental processes and, at the same time, leads to constitutive induction of stress response genes.

## MDF controls the expression of genes involved in cell cycle control during DDR

We next aimed at understanding how the transcriptional changes in *mdf-1* might be associated with the cell division and cell death phenotype in the RAM. The DDR is a very well-studied signaling pathway in plants that, upon DNA damage, leads to cell cycle arrest, cell death, and DNA repair (Yoshiyama et al, 2013b). We therefore analyzed if a similar transcriptional program was constitutively induced in *mdf-1*. To this end, we compared the 1,912 genes reported to undergo transcriptional changes after incubation in the DSB inducing drug zeocin in WT seedlings (Yoshiyama et al, 2020) with the 15,188 DEG genes found in *mdf-1* in comparison to WT at control conditions (Table S10). A significant overlap of 1,353 genes ($P < 2.502 \times 10^{-47}$), which constitute ~71% of the total set of DEG after DNA damage induction, displayed also transcriptional changes in the absence of MDF (Fig 5A and Table S13). Since the transcription factor SOG1 has been shown to coordinate the activation of the plant DDR (Yoshiyama et al, 2013a), we also compared the transcripts that showed either increased IR or that showed mis-expression with log$_2$fold changes above 2 (783 genes) and below −2 (1,557 genes) in *mdf-1* compared with WT, with the set of 146 genes that are known as direct targets of SOG1 upon DNA damage (Ogita et al, 2018). We found a statistically significant

($P < 0.024$) overlap of 29 genes which constituted 20% of the total set of SOG1 target genes (Fig 5B and Table S14). Among them were the transcription factors ANAC044 and ANAC085, which were highly increased with log$_2$fold changes of 2.4 and 5.3, respectively. In addition, *ANAC085* appeared to be also differentially spliced in the absence of MDF (Table S1). Increased expression of ANAC044 and ANAC085 was shown to lead to G2-specific cell cycle arrest by promoting the accumulation of Rep-MYB transcription factors, which negatively control the expression of mitosis-specific genes (Takahashi et al, 2019). To further confirm mis-expression of mitosis-specific genes in *mdf-1*, we compared the set of 279 down-regulated genes in *mdf-1* that were associated with the GO term "cell cycle "(Fig 4C), with the 80 loci that were reported to be Rep-MYB specific target genes after DNA damage induction (Bourbousse et al, 2018) (Fig 5B and Table S15). Indeed, there was a statistically significant ($P < 1.798 \times 10^{-39}$) overlap of 30 genes (38% of Rep-MYB target genes), further confirming that many genes that are mis-expressed in *mdf-1* are involved in DNA damage–induced cell cycle control (Fig 5C). To confirm that the increased expression of *ANAC044* and *ANAC085* and reduced transcription of mitosis-specific genes occurred in an MDF-dependent manner, we performed RT-qPCR analyses on RNA isolated from seedlings of WT, *mdf-1*, *mdf-2*, and *mdf-1::pMDFMDFg*. In both mutant lines, the expression of *ANAC085* and *ANAC044* was significantly increased as compared with WT and the complementation line (Fig 5D). In contrast, as expected, each of the seven mitotic genes tested showed an opposite expression pattern (Fig 5E). The complementation line exhibited increased levels of *MDF* (Fig 5F), which could explain why for some genes (like *SCL28* and *IMK2*), the down-regulation was not only reversed to WT levels but even switched to up-regulation.

We next tested if the cell division and cell death defects of *mdf-1* were caused by ectopic activation of the DNA damage signaling component SOG1 or its upstream regulator ATM. To this end, *mdf-1sog1-7* and *mdf-1atm-2* double-mutants were generated by crossing with *sog1-7* and *atm-2* single-mutants, respectively (Garcia et al, 2003; Sjogren et al, 2015). Analysis of root growth in three dag seedlings revealed that loss of *SOG1* or *ATM* function did not rescue the growth defects of *mdf-1* (Fig 6A). PI staining of root tips showed that the organization of the RAM was unchanged in the *mdf-1sog1-7* as compared with the *mdf-1* single-mutant (Fig 6B), indicating that the cell cycle arrest phenotype occurred also in the absence of ATM or SOG1. To further test if MDF might regulate the expression of proliferation-associated genes downstream or independent of SOG1, we analyzed the expression of *ANAC044* and *ANAC085* by RT-qPCR and found that both transcription factors were significantly increased in the *mdf-1sog1-7* double-mutant to a similar level as in the *mdf-1* single-mutant (Fig 6C). Likewise, analysis of cell death in PI-stained root tips revealed that the size of the cell death area was unchanged in the double-mutant compared with the single-mutant

transcription (F), and DNA repair (G). Intron retention levels were quantified by dividing intron-specific by exon-specific expression using appropriate primers binding specifically to either the intronic or exonic regions of each tested gene. Expression levels were normalized to WT values which were set to one. Captures from the Integrative Genomics Viewer software corresponding to the read coverage tracks for *mdf-1* and WT are represented below each graph. Big boxes represent exons and small boxes represent introns. Intron retention events verified by RT-qPCR are highlighted in orange. Average ± SD of at least two independent biological replicates is represented. *$P < 0.05$; **$P < 0.005$; ***$P < 0.0005$ in comparison to WT as determined by a two-tailed *t* test.

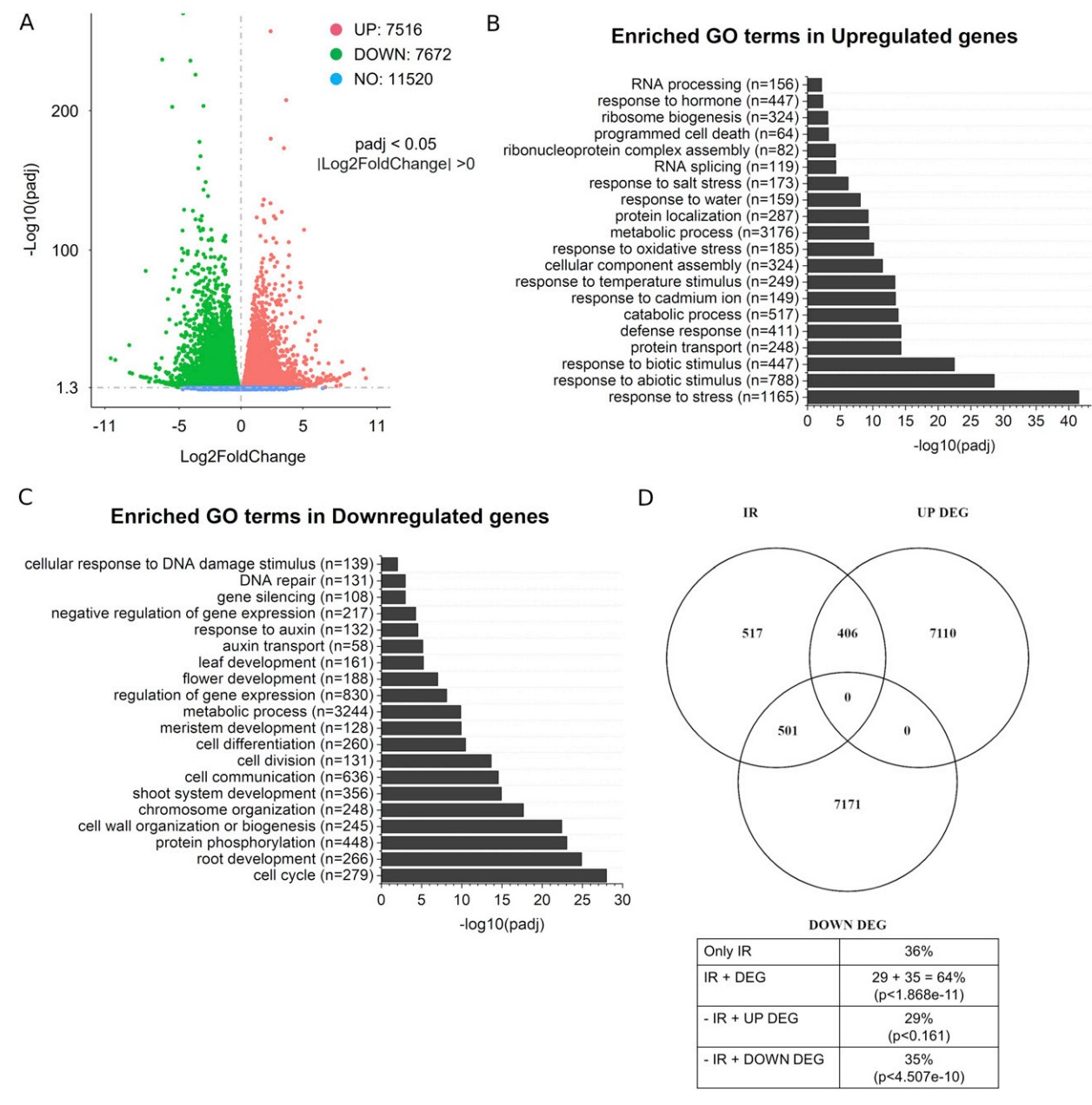

**Figure 4. Loss of *MDF* function leads to altered expression of a large number of genes involved in stress response and cell cycle control.**
**(A)** Volcano plot representing the differentially expressed genes (DEGs) up-regulated (in red) and down-regulated (in green) in *mdf-1* compared with WT with *P*-values < 0.05. Blue dots indicate genes that show no differential expression. **(B)** 20 most representative top significant biological process GO terms enriched in up-regulated genes in *mdf-1* compared with WT. **(C)** 20 most representative top significant biological process GO terms enriched in down-regulated genes in *mdf-1* compared with WT. **(D)** Venn diagram representing the overlap between intron retained and differentially expressed genes and its percentual distribution shown in the table below. Statistical significance of each of the represented overlaps in the Venn diagram was calculated using a normal approximation of a hypergeometric probability formula implemented at the web tool http://nemates.org/MA/progs/overlap_stats.html. *Arabidopsis* reference number of protein-coding genes was set to 27,474.

(Fig 6D). However, the number of seedlings showing cell death was significantly increased in the *mdf-1sog1-7* double-mutant (Fig 6E), indicating that genome instability was further increased in the absence of SOG1. Taken together, these results showed that a transcriptional program, which is normally only induced under stress conditions such as DNA damage, was constitutively activated in *mdf-1* and that this occurred independently of SOG1.

## The phosphorylation state of MDF influences its function during cell division control and pre-mRNA splicing

Our data suggested that MDF was under control conditions important for the proper expression of proliferation-related genes required for cell division activity and growth of meristems. Interestingly, MDF was previously identified among the proteins that became specifically phosphorylated at S22 upon DNA damage

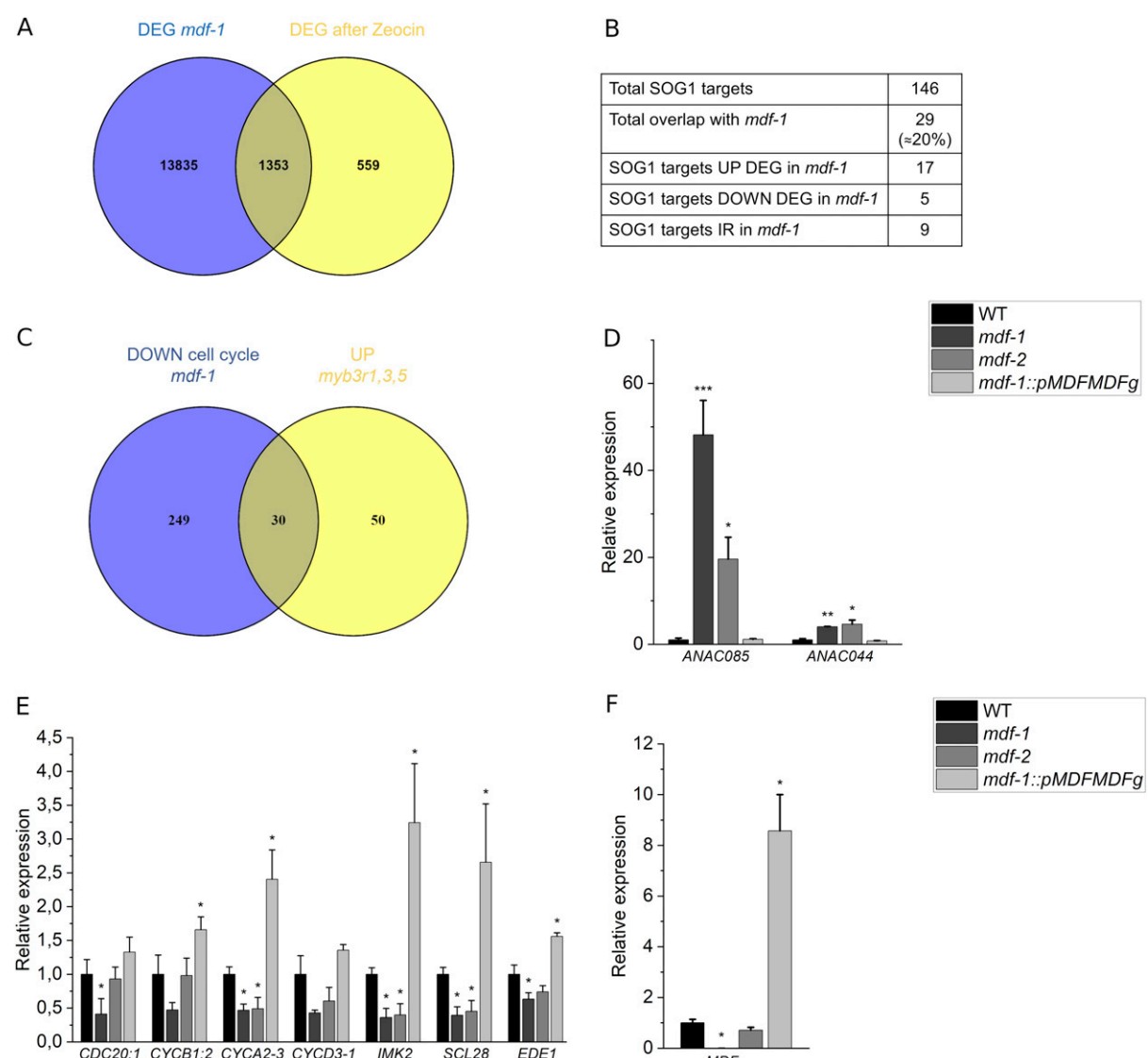

**Figure 5. *MDF* loss of function alters the expression of genes involved in cell cycle control during Arabidopsis DNA damage response.**
**(A)** Venn diagram representing the overlap between differentially expressed genes in *mdf-1* under control conditions and in WT upon treatment with zeocin (Yoshiyama et al, 2020). Statistical significance of the overlap was calculated using a normal approximation of a hypergeometric probability formula implemented at the web tool http://nemates.org/MA/progs/overlap_stats.html. *Arabidopsis thaliana* reference number of protein-coding genes was set to 27,474. **(B)** Quantification of the overlap between direct SOG1 targets and genes showing IR, or a |log₂fold| change value >2 in *mdf-1* background. **(C)** Venn diagram representing the overlap between the 279 down-regulated genes in *mdf-1* associated with the GO biological process "cell cycle" and the 80 loci up-regulated in *myb3r1,3,5* mutant background after DNA damage induction. **(D, E)** Verification of transcriptional changes found by RNA-seq by RT-qPCR analysis in seedlings of WT, *mdf-1*, *mdf-2*, and *mdf-1::pMDFMDFg* on the transcription factors *ANAC085* and *ANAC044* (D) and mitosis-related genes (E). **(F)** *MDF* expression measured by RT-qPCR in seedlings of WT, *mdf-1*, *mdf-2*, and *mdf-1::pMDFMDFg*. Average ± SD of at least two independent biological replicates is shown. *$P < 0.05$; **$P < 0.005$; ***$P < 0.0005$ in comparison to WT as determined by a two-tailed *t* test.

(Roitinger et al, 2015). To test if this phosphorylation influenced MDF activity, we used the MDF cDNA to generate two constructs in which either the S22 residue remained unchanged (35S::MDFYFP) or in which it was changed to alanine (35S::MDFS22AYFP) and thus could not be phosphorylated anymore. Both constructs were placed under the control of a constitutive promoter (CAMV35S), transformed into the *mdf-1* mutant, and stably transformed lines were established. We confirmed that both constructs were expressed at similar levels in the individual transgenic lines by measuring transcript abundance by RT-qPCR (Fig 7A) and protein accumulation

by imaging of MDF-YFP–associated fluorescence in roots (Fig 7B). Analysis of root development revealed that both constructs resulted in a partial recovery of root growth. However, root length was significantly increased in the *mdf-1::p35SMDF* lines as compared with *mdf-1::p35SMDFS22A* plants (Figs 7C and S4B). We confirmed that the increased growth was because of increased cell division rates by assessing the number of dividing cells in the cortical cell layer of the RAM (Figs 7D and S4A). PI staining of root tips showed that in transgenic lines, for both constructs, dead cells accumulated in the meristematic zone. However, the area of cell

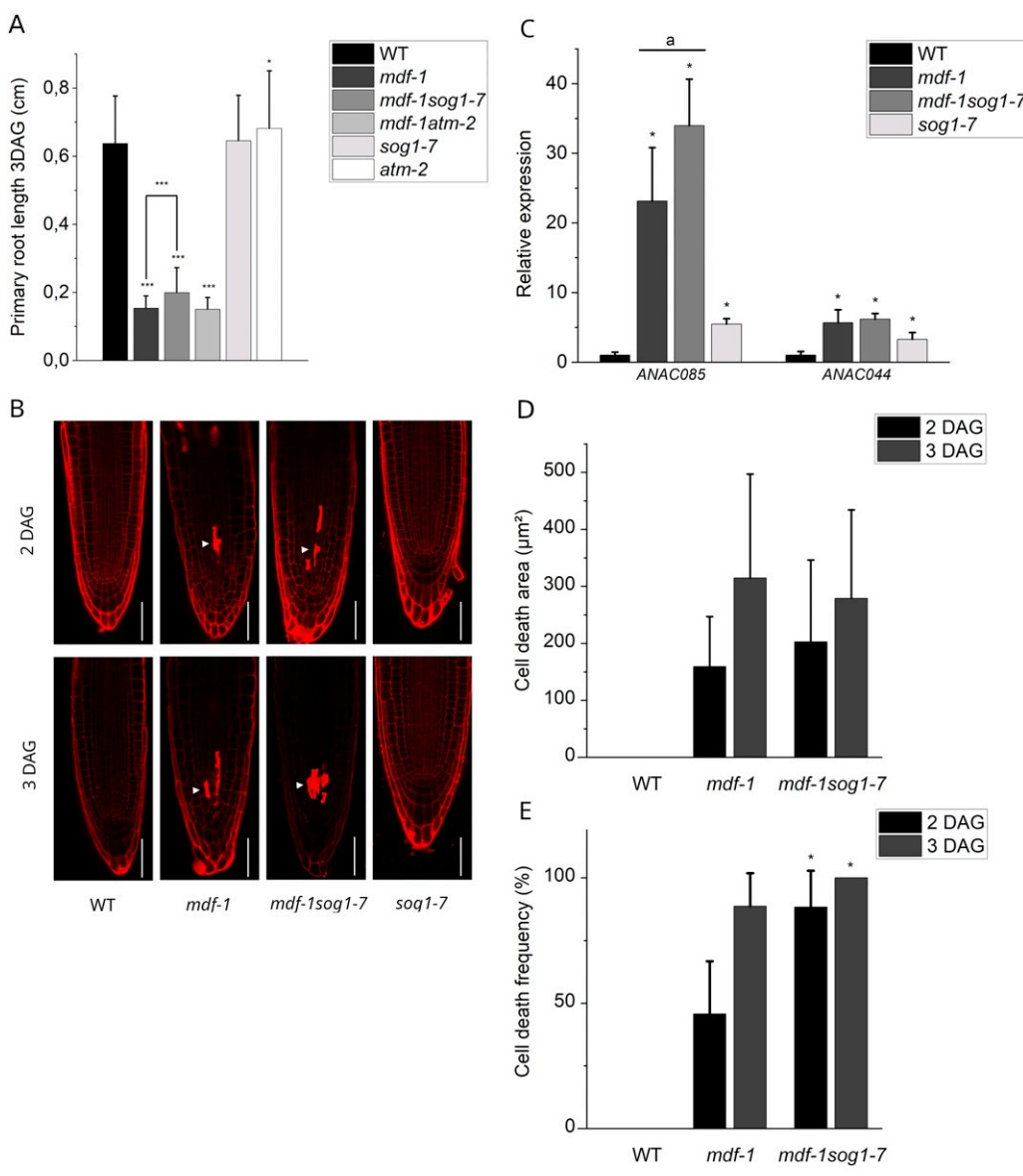

**Figure 6. Root growth arrest in *mdf-1* occurs independently of SOG1 and ATM.**
**(A)** Primary root length of 3 days after germination (dag) seedlings of WT (n = 110), *mdf-1* (n = 57), *mdf-1sog1-7* (n = 45), *mdf-1atm-2* (n = 38), *sog1-7* (n = 74), and *atm-2* (n = 146) plants. Statistical significance was determined in comparison to WT. **(B)** Representative confocal images of PI-stained root tips of WT, *sog1-7*, *mdf-1*, and *mdf-1sog1-7* in 2 and 3 dag seedlings. Arrowhead marks dead cells. Scale bar: 50 µm. **(C)** RT-qPCR analysis of 12-d-old WT, *mdf-1*, *mdf-1sog1-7*, and *sog1-7* seedlings on *ANAC085* and *ANAC044*. Average ± SD of three independent biological replicates is represented. *$P < 0.05$ as determined in comparison to WT by a one-way ANOVA post hoc Tukey–Kramer test. "a" represents statistical significance in comparison to *sog1-7* as determined by a one-way ANOVA post hoc Tukey–Kramer test. **(D, E)** Quantification of the cell death area in root tips (D) and of the frequency of seedlings showing cell death (E) in WT (n = 21 and n = 22), *mdf-1* (n = 21 and n = 95), and *mdf-1sog1-7* (n = 32 and n = 25) lines at 2 and 3 dag. Statistical significance was determined in comparison to *mdf-1*. Average ± SD is represented. *$P < 0.05$; **$P < 0.005$; ***$P < 0.0005$ as determined by a two-tailed *t* test.

death was significantly increased in *mdf-1::p35SMDFS22A* seedlings (Fig 7B and E). These phenotypic differences demonstrated that the phosphorylation status of MDF indeed influenced its ability to rescue the mutant phenotype and thus seemed to be critical for MDF activity. To assess if this was associated with splicing defects, we performed RT-qPCR experiments to analyze IR in transcripts that were found to be mis-spliced in *mdf-1* (Fig 3). Indeed, the IR defect was for some genes, such as *CYCB2-2* and *RAD51D*, reversed in both lines, whereas for others, such as in *MYB3R3*, it was only rescued in

the *mdf-1::p35SMDF* line but not in *mdf-1::p35SMDFS22A* (Fig 7F). Before MYB3R3 is involved in the repression of mitotic genes, the transcript levels of two mitotic genes down-regulated in *mdf-1* background (Fig 5D) were also tested. Although the expression of *CDC20;1* was restored to WT levels in both transgenic lines, the expression of *CYCA2-3* was restored to WT levels only in *mdf-1::p35SMDF* seedlings but not in *mdf-1::p35SMDFS22A* seedlings (Fig 7G). These results suggested that the activity of MDF might indeed be controlled by its phosphorylation status.

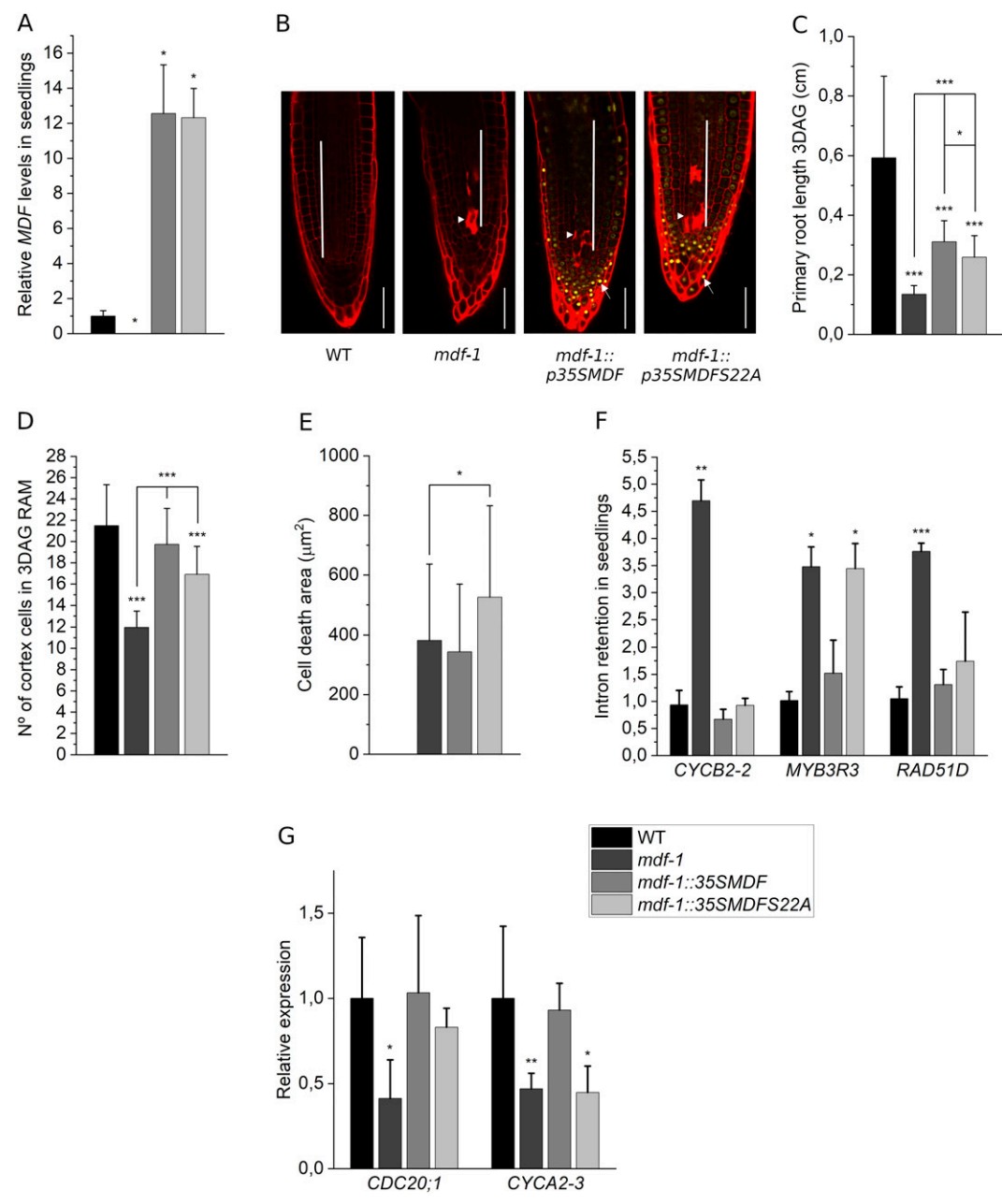

**Figure 7. Phosphorylation state of MDF influences its function during cell division control and pre-mRNA splicing.**
**(A)** *MDF* expression by RT-qPCR in WT, *mdf-1*, *mdf-1:p35SMDF*, and *mdf-1:p35SMDFS22A* 12 days after germination (dag) seedlings (n = 3). Statistical significance was determined in comparison to WT. **(B)** Representative confocal images of PI-stained root tips of WT, *mdf-1*, *mdf-1::p35SMDF*, and *mdf-1::p35SMDFS22A* 3 dag seedlings. Arrowhead marks dead cells. Arrow marks nuclei showing YFP associated fluorescence. White bar indicates the division zone. Scale bar: 50 μm. **(C)** Primary root length of 3 dag seedlings of WT (n = 25), *mdf-1* (n = 15), *mdf-1::p35SMDF* (n = 19), and *mdf-1::p35SMDFS22A* (n = 17). Statistical significance was determined in comparison to WT and *mdf-1*. **(D)** Number of dividing cells in the cortical layer of 3 dag seedlings of WT (n = 40), *mdf-1* (n = 36), *mdf-1::p35SMDF* (n = 10), and *mdf-1::p35SMDFS22A* (n = 10). Statistical significance was determined in comparison to WT and *mdf-1*. **(E)** Quantification of cell death area in 3 dag seedlings of WT (n = 24), *mdf-1* (n = 35), *mdf-1::p35SMDF* (n = 29), and *mdf-1::p35SMDFS22A* (n = 19). Statistical significance was determined in comparison to *mdf-1*. **(F, G)** RT-qPCR analysis to confirm increased IR (F) and down-regulation of mitotic genes (G) found by RNA-seq in samples of 12 dag seedlings of WT, *mdf-1*, *mdf-1::p35SMDF*, and *mdf-1::p35SMDFS22A* (n = 2–3). Statistical significance was determined in comparison to WT. Average ± SD is represented. *P < 0.05; **P < 0.005; ***P < 0.0005 as determined by a two-tailed *t* test.

## Loss of conserved plant splicing factors influences DDR

We next aimed at understanding how impaired activity of con-served plant splicing factors might influence DDR signaling. To this end, we tested the sensitivity of mutant lines for three splicing factors to the double-strand breaks inducing drug zeocin. These were *mdf-1* and *mdf-2*, as well as T-DNA insertion lines for the core splicing factor LSM8 and the SR-like protein SR45, which both were reported to be involved in developmental processes and abiotic stress response (Ali et al, 2007; Perea-Resa et al, 2012; Carrasco-

Lopez et al, 2017). Seedlings were germinated on MS medium and transferred after 5 d to plates supplemented with zeocin. Measurement of root length at 2, 4, and 6 d after transfer (DAT) revealed that in each of the lines root growth was reduced in the presence of zeocin (Fig 8A). Compared with WT, mdf-1, lsm8-1, and sr45-1 were significantly more affected by presenting complete growth arrest already 4 d after transfer to zeocin supplemented media (Fig 8B). Next, we analyzed the impact of zeocin on shoot development by growing seedlings on control medium or medium supplemented with two different concentrations of zeocin. Quantification of the seedlings that formed the first true leaf pair after 10 d of growth revealed that mdf-1, mdf-2, lsm8-1, and sr45-1 were significantly more affected than WT. Almost no mutant seedling was able to develop true leaves when grown in the highest zeocin concentration (Fig 8C). These results indicated that mdf-1, mdf-2, sr45-1, and lsm8-1 were hypersensitive to DNA damage treatment. To test whether this was because of an altered transcriptional response to DNA damage, seedlings were incubated for 2 h with zeocin and used for RT-qPCR analysis of the four DDR genes: BREAST CANCER SUSCEPTIBILITY1 (BRCA1), RAD51, POLY (ADP-RIBOSE)-POLYMERASE1 (PARP1), and POLY (ADP-RIBOSE)-POLYMERASE2 (PARP2). Although all lines showed an increased expression of each of these genes upon zeocin treatment, the level of induction was much lower in mdf-1, mdf-2, and sr45-1 than in WT and lsm8-1 (Fig 8D). PI staining of root tips revealed in each of the lines, including WT, a robust induction of cell death upon zeocin treatment (Fig 8E). Moreover, this analysis revealed that the sr45-1 mutant exhibited even under control conditions accumulation of dead cells in the division zone of the RAM in a similar pattern as mdf-1 and mdf-2 (Fig 8F). Together, these results indicated that MDF and SR45 seem to be important for DDR signaling and maintenance of genome stability. To test if DNA damage might influence MDF activity by altering its subcellular localization pattern, we analyzed MDF-GFP or MDF-YFP associated fluorescence in root tips of the complemented line (mdf-1::pMDFMDFg) and of mdf-1::p35SMDF and mdf-1::p35SMDFS22A at control conditions and after treatment with zeocin by confocal microscopy. However, no changes between control and zeocin treated samples were observed (Fig S5). Therefore, the mechanism by which MDF might influence DDR is still unknown.

## Discussion

MDF was previously identified as an important factor for meristem development and proposed to control the expression of patterning genes (Casson et al, 2009). Here, we show that MDF is associated with a core component of the spliceosome and important for correct splicing of numerous transcripts in young seedlings. MDF loss of function was associated with constitutive activation of a similar transcriptional program that is in WT induced upon DNA damage. Our analysis of a dephosphomutant version of MDF indicated that the MDF phosphorylation state was important for its function in maintaining meristem activity. Moreover, we show that additional well-established mutants for plant splicing proteins are hypersensitive to DNA damage

treatment, underscoring the importance of alternative splicing mechanisms during plant DDR.

Since in plants a suitable in vitro plant-splicing system is so far not established, the individual components of the plant spliceosome were identified based on sequence similarity to human and yeast proteins (Albaqami et al, 2019). MDF is known as a SART1-like protein that shares 38.9% and 25.9% overall sequence similarity with its human and yeast homologs hSART1 and ySNU66, respectively. We found that MDF interacted with the U6 snRNP–bound protein LSM8 and physically interacts with the U4/U6.U5 tri-snRNP component STA1. Consistent with this, human SART1 and yeast SNU66 co-purified with the U4/U6.U5 tri-snRNP complex. Y2H-based interaction studies revealed that SART1 physically interacted with the U5-associated proteins hBRR2A and hPERP6 and the U4/U6–associated protein hPRP3, whereas in the yeast spliceosome, SNU66 interacted only with the two U5-specific yPRP8 and yBRR2 (Nguyen et al, 2016). Although SART1 was suggested to act as a bridging protein between the U5 and U4/U6 snRNPs, it was found to be not essential for the assembly or the stability of the tri-snRNP, and therefore, its exact function within the spliceosome still remains to be elucidated (Liu et al, 2006). We found that MDF co-localized with the well-studied plant splicing protein SR45 in the nucleoplasm and in nuclear speckles, and our results are in line with data from human cells showing that SART1 accumulates in the nucleoplasm, in nuclear speckles, and also in Cajal bodies in which tri-snRNP assembly occurs (Yildirim et al, 2021). It will be interesting to identify in future experiments all physical interaction partners of MDF within the plant spliceosome and to test whether it might function during tri-snRNP assembly or stability.

We found that under control conditions, the mdf mutants display a phenotype characterized by decreased cell division rates, increased endoreduplication level, and accumulation of dead cells in the division zone of the RAM (Figs 1 and S1). In human cells, increased expression of SART1 in different cancer cell lines leads to cell division defects and apoptosis and correlates with altered expression of important cell cycle regulators (Hosokawa et al, 2005). SART1 was also shown to be essential for cell division in breast cancer cells (Kittler et al, 2004), and siRNA-mediated down-regulation of SART1 resulted in increased apoptosis (Allen et al, 2012). However, how these cell division related defects might be associated with altered pre-mRNA splicing is still not completely understood. The MDF phenotype resembles the phenotype of WT seedlings after induction of DNA damage, and accordingly, it also shows altered expression of genes regulated by DNA damage in plants (Yi et al, 2014; Ogita et al, 2018). Among the genes that are highly mis-expressed in the mdf mutants are the transcription factors ANAC044 and ANAC085 (Fig 5C), whose activation upon DNA damage promotes the stabilization of the MYB3R transcriptional repressors, resulting in reduced expression of G2/M genes and cell cycle arrest (Takahashi et al, 2019). The importance of splicing factors in cell cycle progression in plants has been suggested in a recent study in which the role of PLEIOTROPIC REGULATORY LOCUS 1 (PRL1), a conserved regulator of splicing in plants (Wang et al, 2021), was analyzed. Loss of PRL1 leads to IR of important cell cycle genes including the CYCD1;1 and CYCD3;1 and the prl1 mutant exhibits delayed cell cycle progression and increased resistance to the replication stress inducing drug hydroxyurea (Wang et al, 2021).

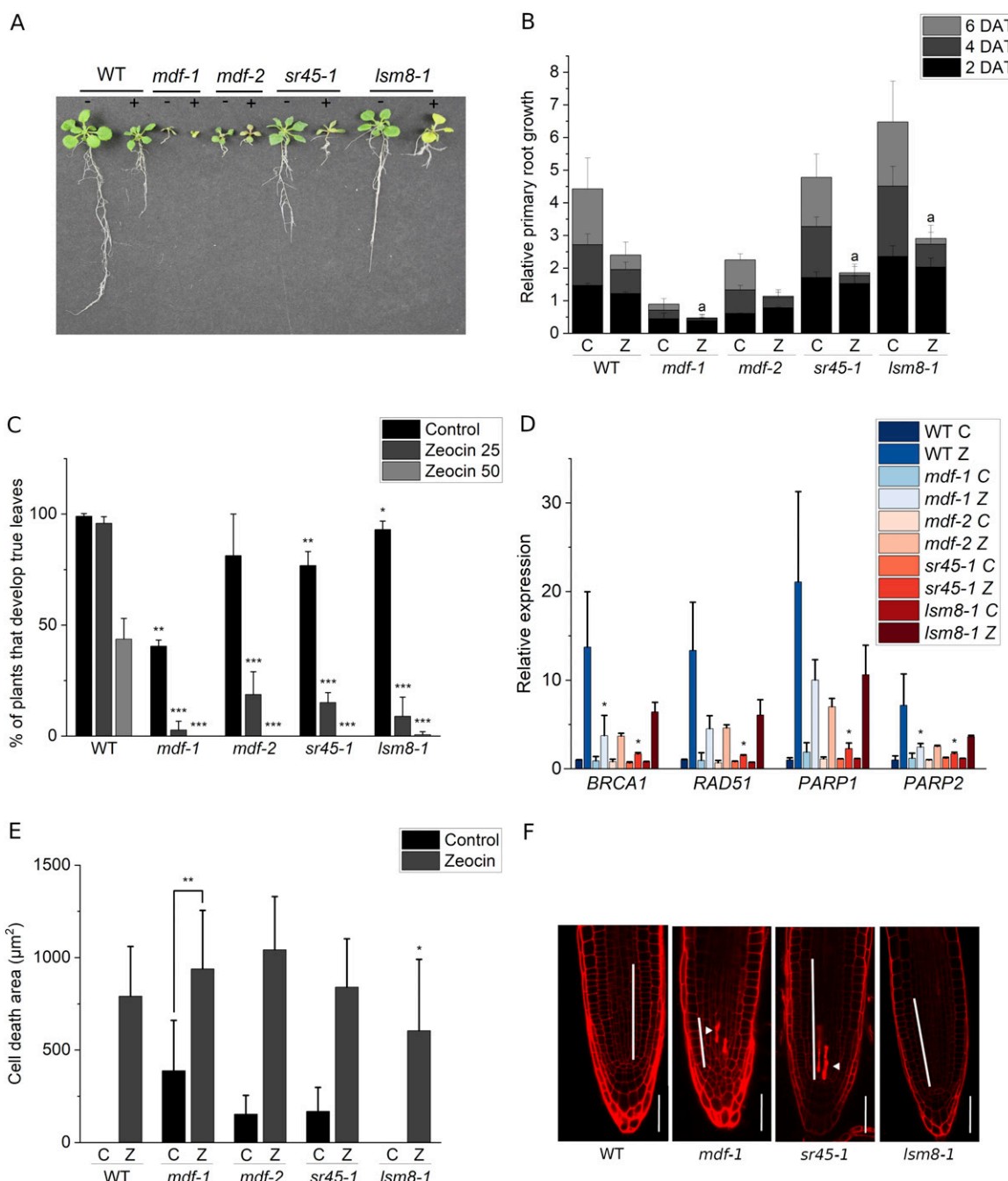

**Figure 8. Mutant lines for *MDF*, *LSM8*, and *SR45* exhibit increased sensitivity to zeocin treatment.**
**(A)** Representative photograph of WT, *mdf-1*, *mdf-2*, *sr45-1*, and *lsm8-1* seedlings at 18 days after germination (dag). At 4 dag, seedlings were transferred to MS medium (−) and MS medium supplemented with 20 μg/ml zeocin (+). **(B)** Quantification of relative primary root growth of seedlings of WT, *mdf-1*, *mdf-2*, *sr45-1*, and *lsm8-1*. **(C)** Root length was measured at 3 dag before transfer and at 2, 4, and 6 d after transfer (dat) to MS medium (C) or medium supplemented with 20 μg/ml zeocin (Z). The panel shows ratio of root length from 2, 4, and 6 dat/3 dag for each line measured in at least three independent experiments (a minimal of 20 plants per line, condition, and experiment were measured). "a" represents no statistical difference between primary root length at 4 and 6 d after transfer according to a two-tailed *t* test. **(C)** Percentage of plants that developed true leaves at 10 dag grown on control conditions (Control) or on medium containing zeocin 25 μg/ml (zeocin 25) or zeocin 50 μg/ml (zeocin 50) for WT, *mdf-1*, *mdf-2*, *sr45-1*, and *lsm8-1*. The panel shows results from at least three independent experiments (at least 20 plants per line, condition, and experiment) *P < 0.05; **P < 0.005; ***P < 0.0005 as determined in comparison to WT by a two-tailed *t* test. **(C, D)** RT-qPCR analysis showing induction of DNA repair genes in two biological replicates of six dag seedlings of WT, *mdf-1*, *mdf-2*, *sr45-1*, and *lsm8-1* grown at control conditions (C) and after transfer to 100 μM zeocin containing medium for 2 h (Z). *P < 0.05 as determined in comparison to WT by a one-way ANOVA post hoc Tukey–Kramer test. **(C, E)** Quantification of cell death area in seedlings that were at two dag incubated for 24 h in MS medium (C) or in medium containing 15 μM zeocin (Z) for WT (n = 38 and n = 39), *mdf-1* (n = 32 and n = 35), *mdf-2* (n = 31 and n = 38), *sr45-1* (n = 31 and n = 30), and *lsm8-1* (n = 32 and n = 35). *P < 0.05; **P < 0.005; ***P < 0.0005 as determined in comparison to WT by a two-tailed *t* test. Average ± SD is shown. **(F)** Representative confocal images of PI-stained root tips of three dag seedlings of WT, *mdf-1*, *sr45-1*, and *lsm8-1* grown on MS medium. Arrowhead marks dead cells. White bar indicates the division zone. Scale bar: 50 μm.

PRL1 was proposed to be a constitutive positive regulator of cell division that is targeted for degradation upon replication stress (Wang et al, 2021). MDF seems to have a similar function in promoting cell cycle progression by regulating splicing and expression of proliferation-related genes in meristems.

In addition, *mdf* mutants exhibit accumulation of dead cells in the root meristem and a disorganized patterning of the RAM (Fig 1B). A link between pre-mRNA splicing and maintenance of the root stem cell niche was also demonstrated by the analysis of the *rdm16-4* mutant (Lv et al, 2021). RDM16 encodes the core splicing protein PRP3 that is necessary for the assembly of the U4/U6 snRNP (Wan et al, 2016). The *rdm16-4* mutant showed impaired root meristem development and mis-splicing of genes involved in cytokinin signaling and meristem patterning, such as the transcription factors of the PLETHORA family, resulting in reduced cell division rates (Lv et al, 2021). Thus, *rdm16-4* and *mdf-1* mutants show to some extent similar defects in root development. However, the absence of MDF also affects shoot development (Fig 1E), and therefore, MDF has additional functions necessary for plant development and survival, which might be associated with the regulation of stress-induced genes.

We analyzed the overlap of differentially expressed genes in WT upon zeocin treatment and in the *mdf-1* mutant at control conditions and found that there is a significant overlap of 1,353 genes. This result suggests that in the *mdf-1* mutant, response to DNA stress is constitutively activated. Our DDR response assays revealed that upon zeocin treatment, the *mdf-1* and *mdf-2* mutant seedlings showed a hypersensitive phenotype. However, both mutant lines were able to induce the expression of DNA repair genes, although to a lower level than the WT (Fig 8B and C). Our current hypothesis is that MDF is important for safeguarding genome stability in dividing tissues by ensuring proper expression of cell cycle genes at control conditions. In addition, upon DNA stress, it might be involved in the coordination of cell cycle gene expression with the efficient induction of DNA repair genes.

Interestingly, SR45 is homologous to animal RNA BINDING PROTEIN S1, which is implicated in cell cycle control and maintenance of genome stability (Li et al, 2007; Fukumura et al, 2018). The *sr45-1* mutant of *Arabidopsis* was previously shown to exhibit developmental defects including delayed root growth and altered splicing of important genes for plant development (Ali et al, 2007). Our finding that the *sr45-1* mutant accumulates dead cells in the RAM under control conditions and shows decreased induction of DNA repair genes upon DNA damage treatment indicates a so far unknown function for this important splicing regulator in plant DDR. However, despite the constitutive accumulation of dead cells in the RAM, *sr45-1* showed a similar induction of cell death upon DNA damage as WT. In contrast, in *mdf-1,* the relative increase of dead cells upon DNA damage is diminished (Fig 8E). It will be interesting to test in future experiments if MDF—similar to its human homolog SART1—is directly involved in cell death activation in dividing cells.

Our genome-wide analysis of pre-mRNA splicing defects revealed that 1,894 IR events occurred in *mdf-1* compared with WT. By comparing the list of 1,521 genes showing IR in *mdf-1* with those that were mis-spliced in previously published mutants for three other components of the plant tri-snRNP, we found that about 50% of IR target genes were only mis-spliced in *mdf-1* but not affected in

any of the other three mutants tested. Interestingly, those targets that were only mis-spliced in *mdf-1* were enriched for genes involved in stress response and RNA or DNA metabolism. This could indicate that correct splicing of these MDF-specific targets can still occur in the absence of either of the three other splicing proteins but not in the absence of MDF. Whether MDF is itself involved in targeting the spliceosome to these targets or whether this is a down-stream effect caused by mis-splicing or mis-expression of other spliceosome proteins still needs to be resolved.

It was previously shown that the phosphorylation of MDF at S22 increases in seedlings upon DNA damage treatment (Roitinger et al, 2015). We performed complementation experiments using a dephosphomutant version of MDF to test if the phosphorylation at S22 was important for MDF function. Indeed, we found that the dephosphomutant version of MDF was unable to fully complement the cell division phenotype of *mdf-1* (Fig 7).

Splicing-related proteins have been previously identified as major phosphorylation targets in plants (de la Fuente van Bentem et al, 2006). Phosphorylation of SR proteins in their RS domain was shown to alter their activity by influencing their subcellular location (Huang & Steitz, 2005). Moreover, the pre-mRNA processing 4 KINASE A, whose loss of function was associated with several developmental defects such as late flowering, reduced branching, and lowered seed set (Kanno et al, 2018), was proposed to influence splicing patterns by phosphorylating a subset of splicing regulators (Kanno et al, 2018). A direct link between stress response, phosphorylation, and splicing was recently established by the analysis of mitogen-activated protein kinase 4 (MPK4), a major activator of the plant immune response. The absence of MPK4 leads to altered splicing of several splicing factors and immunity-related protein kinases (Bazin et al, 2020). Our results provide a first insight into the mechanism by which MDF splicing activity might be controlled. However, further experiments are required to find out at a genome-wide level how the MDF phosphorylation state affects the splicing patterns in dividing tissues and how MDF phosphorylation is regulated.

Together, our data suggest the following model: Under control conditions, MDF would promote the expression and correct splicing of genes maintaining cell division and growth in meristematic tissues. When MDF is inactive, a transcriptional program that is normally only induced under stress conditions would then become activated, and this results in cell cycle arrest, induction of endoreplication, and cell death.

## Materials and Methods

### Plant material and growth conditions

Wild type for all experiments was the *Arabidopsis thaliana* accession Columbia (Col-0), and every mutant line used during this study was in Col-0 background. T-DNA insertion lines for *mdf-1* (SALK_040710), *mdf-2* (SAIL_775_F10), *sr45-1* (SALK_004132), *lsm8-1* (SALK_025064), and *atm-2* (SALK_006953) and the mutant line *sog1-7* were previously described (Garcia et al, 2003; Ali et al, 2007; Casson et al, 2009; Perea-Resa et al, 2012; Sjogren et al, 2015). Double-mutants for *mdf-1sog1-7* and *mdf-1atm-2* were generated by crossing. The complementation line *mdf-1::pMDFMDFg* was achieved by floral dipping

(Clough & Bent, 1998) using *Agrobacterium tumefaciens* C58C1, harboring full genomic MDF and promoter in the pMDC107 destination vector (Curtis & Grossniklaus, 2003). Primers for fragment amplification were described (Casson et al, 2009) and used for cloning inside the pDONR221 vector (Thermo Fisher Scientific). The transgenic plants overexpressing MDF full-length coding sequence (CDS) were generated by amplifying MDF full-length CDS, excluding the STOP codon with primers described in Table S16 for cloning inside the pENTR-D-TOPO plasmid (Thermo Fisher Scientific) followed by LR recombination reaction with the destination vector pEG101. The final step was floral dipping using the *A. tumefaciens* C58C1 strain. The making of the MDFS22ACDS construct involved an additional step before LR reaction in which serine 22 was exchanged to alanine by site-directed mutagenesis PCR using appropriate overlapping primers harboring the mutation and MDFCDS_pENTR-D-TOPO as template. The PCR product was transformed in *Escherichia coli* TOP10 with previous DpnI incubation to avoid transformation of non-mutated plasmid. Site-directed mutagenesis primers can be found in Table S16. For in vitro analyses, plants were grown either on soil or on Murashige and Skoog (MS) plates containing 1% sucrose and 1% (wt/vol) agar in growth chambers (16 h light, 22°C/8 h dark, 18°C cycles). Seeds were vapor sterilized by a mixture of 2.3 ml of 32% HCl and 50 ml of NaOCl for 3 h.

**Growth assays and propidium iodide and FDA staining**

Primary root growth was assessed by germination and growth on MS vertical plates. Plates were scanned at the indicated timepoints. Quantification was achieved using the ImageJ software. Zeocin root growth inhibition experiments were carried out by germinating plants on MS vertical plates and transferring them to control (no zeocin) and zeocin (Invitrogen) 20 $\mu$g/ml for 6 additional days. Scanning was performed before transfer (after 4 d in light) and 2, 4, and 6 d after transfer. Primary root length was measured at each timepoint, and growth for each timepoint was determined by subtracting the length of the previous timepoint. Relative growth values were obtained by dividing the lengths of each timepoint by the initial root length measured before transfer. For true leaf formation experiments, plants were germinated and grown on horizontal plates containing no zeocin (Control), zeocin 25 $\mu$g/ml (zeocin 25), and zeocin 50 $\mu$g/ml (zeocin 50) for 10 d. Percentage of plants showing true leaves was calculated for each of the conditions in at least three independent experiments with a minimum of 20 plants per line, condition, and experiment. Propidium iodide (PI) (Sigma-Aldrich) staining of 2 and 3 dag plants was performed by cutting the root tips of the plants at the desired age and incubating them for 1 min in a 10 $\mu$g/ml PI solution. For FDA staining, FDA (Cat. no. F7378; Sigma-Aldrich) was dissolved in acetone to a stock concentration of 5 mg/ml. The stock solution was diluted 1:100 in $H_2O$, and seedlings were incubated for 20 min and washed three times in $H_2O$ before imaging. Imaging was carried out using confocal microscopy with layer specification based on the visualization of the quiescent center. The cell death area occupied by the PI-stained cells was quantified using the ImageJ software. For induction of cell death analyses, before PI staining, plants were transferred to liquid MS medium with or without zeocin at a concentration of 15 $\mu$M overnight. The number of dividing cells in the RAM was determined by counting the amount of cortex cells within the dividing zone of the RAM. Layer specification was set based on the visualization of the quiescent center, and the division zone was determined as the population of small and round stem cells and progenitors located underneath the first elongated cell.

**Flow cytometry**

Flow cytometry was performed as described previously (Kallai et al, 2020). Nuclei were isolated from seedlings (Partec CyStain UV precise P kit) and analyzed by flow cytometry on BD LSRII (BD Biosciences) with a solid-state laser (Ex 405 nm) and 450/50 band pass filter. Data evaluation was performed in FlowJo from at least three independent experiments. The endoreplication index was determined from percentage values of each C-level.

**Immunofluorescence labeling**

Slide preparation of squashed *Arabidopsis* roots and immuno-labelling were performed as described in Horvath et al (2017).The anti-$\alpha$-tubulin (1:1,000; Abcam) and anti-mouse Alexa Fluor 488 (1:600) antibodies were used to visualize mitotic microtubular arrays. Chromatin was counterstained with DAPI. Microscopical analyses were performed using FV10 ASW2.0 and confocal IX-81 FV-1000 Olympus microscopes.

**Co-immunoprecipitation experiments**

15 g of 3-wk-old c-lsm8 plants (Perea-Resa et al, 2012) was cross-linked with 1% formaldehyde in PBS, two times for 10 min by vacuum infiltration, followed by 5 min vacuum with glycine to a final concentration of 125 mM. Plant material was rinsed six times with precooled water and frozen in liquid nitrogen. Nuclei isolation was performed as reported previously (Locascio et al, 2013), and LSM8-GFP was immunoprecipitated using the GFP-Trap agarose system (ChromoTek) following the manufacturing indications. SDS–PAGE (10% polyacrylamide) was run till the whole proteome had penetrated in the resolving gel (about 1 cm of total migration). Gels were stained with the Colloidal Blue Staining Kit (Invitrogen). Each proteome was excised and divided in two fractions ("up" and "down"). These fractions were cut in small pieces before manual in-gel digestion with trypsin. Excised bands were separately destained with 50 mM ammonium bicarbonate (ABC) (Sigma-Aldrich) and 50% acetonitrile (ACN) (Fisher Chemical). Samples were then reduced with 10 mM dithiothreitol (Bio-Rad) in 50 mM ABC and alkylated with 55 mM iodoacetamide (GE Healthcare Life Sciences) in 50 mM ABC. Then, gel pieces were digested with porcine trypsin (Thermo Fisher Scientific), at a final concentration 12.5 ng/ml in 50 mM ABC, overnight at 37°C. Peptides were extracted using 100% ACN and 0.5% trifluoroacetic acid (Sigma-Aldrich), purified using a Zip Tip (Millipore, Sigma-Aldrich), and dried (3). Finally, samples were reconstituted in 10 $\mu$l of 0.1% formic acid before their analysis by nanosystem liquid chromatography–tandem mass spectrometry (nLC–MS/MS). All peptide separations were carried out on an Easy-nLC 1000 nanosystem (Thermo Fisher Scientific). For each analysis, the sample was loaded into a precolumn Acclaim PepMap 100 (Thermo Fisher Scientific) and eluted in a RSLC PepMap C18, 15 cm long, 50 $\mu$m inner diameter, and 2 $\mu$m particle size (Thermo Fisher

Scientific). The mobile phase flow rate was 300 nl/min using 0.1% formic acid in water (solvent A) and 0.1% formic acid and 100% acetonitrile (solvent B). The gradient profile was set as follows: 0–35% solvent B for 90 min, 35–100% solvent B for 4 min, 100% solvent B for 8 min. 4 $\mu$l of each sample was injected. MS analysis was performed using a Q Exactive mass spectrometer (Thermo Fisher Scientific). For ionization, 2,000 V of liquid junction voltage and 270°C capillary temperature were used. The full scan method employed a m/z 400–1,500 mass selection, an Orbitrap resolution of 70,000 (at m/z 200), a target automatic gain control (AGC) value of $3 \times 10^6$, and maximum injection times of 100 ms. After the survey scan, the 15 most intense precursor ions were selected for MS/MS fragmentation. Fragmentation was performed with a normalized collision energy of 27 eV and MS/MS scans were acquired with a starting mass of m/z 100, AGC target was $2 \times 10^5$, resolution of 17,500 (at m/z 200), intensity threshold of $8 \times 10^3$, isolation window of 2 m/z units, and maximum IT was 100 ms. Charge state screening was enabled to reject unassigned, singly charged, and equal or more than seven protonated ions. A dynamic exclusion time of 20 s was used to discriminate against previously selected ions. MS data were analyzed with Proteome Discoverer (version 1.4.1.14) (Thermo Fisher Scientific) using standardized workflows. Mass spectra *.raw files were searched against Swiss-Prot *A. thaliana* (thale cress) database (14,986 sequences protein entries) using the Mascot (version 2.6.0, Matrix Science) search engine. Precursor and fragment mass tolerance were set to 10 ppm and 0.02 D, respectively, allowing two missed cleavages, carbamidomethylation of cysteines as a fixed modification and methionine oxidation as a variable modification. Identified peptides were filtered using the Percolator algorithm (Kall et al, 2007) with a q-value threshold of 0.01.

### BiFC, co-localization, and FLIM FRET analyses

For BiFC assays, the CDSs of *MDF*, *LSM8*, and *LSM1a* were amplified from a pool of *Arabidopsis* cDNAs with appropriated primers, cloned into the pDONR207 vector, and then transferred to the pYFN43 and pYFC43 destination vectors for BiFC assays. These constructs were used to transform cells of the *A. tumefaciens* strain GV3101. Transient expression of fusion proteins for BiFC was analyzed by confocal microscopy, 3 d after agroinfiltration in leaves of 3-wk-old *Nicotiana benthamiana* plants grown at 23°C. Analyses were performed at least in triplicate with independent samples. The constructs for FLIM-FRET assays and co-localization experiments were generated by amplifying the CDS excluding the STOP codon of each of the splicing factors by PCR and subsequent cloning into pENTR-D-TOPO for MDF and pDONR221 for LSM2, LSM8, and SR45, followed by LR recombination reaction with the destination vectors pABindGFP and pABindmCherry (Bleckmann et al, 2010). Constructs were transformed in *A. tumefaciens* C58C1 and transiently expressed in 4–6 wk-old *N. benthamiana* leaves. Fluorescence lifetime was acquired with a Leica TCS SP8 confocal microscope (40× water immersion objective). Time-correlated single photon counting was performed with picosecond resolution (PicoQuant Hydra Harp 400). Fluorophores were excited with a 470 nm (r LDHPC470B, 40 MHz) or 485 nm (LDH-D-C-485, 32 MHz) pulsed polarized diode laser with a power of 1 $\mu$W at the objective lens. For detection of emitted light, a SMD-adjusted hybrid detector (HyD SMD3) (wavelength set to 500–520 nm) and a TCSPC module

PicoHarp 300 (PicoQuant) were used. Image acquisition was performed at zoom 6 with a resolution of 256 × 256 pixel with a dwell time of 20 $\mu$s, and photons were collected over 50–60 frames. Fluorescence decay was analyzed in SymPhoTime 64 (version 2.4; PicoQuant) using the Lifetime FRET image analysis tool. TCSPC channels were binned by eight; count threshold was set so that the background was removed. Fluorescence decay was fitted using a multi-exponential decay, and the amplitude-weighted lifetime was considered as the sample's apparent lifetime. FRET efficiency was calculated as the lifetime of the FRET sample over the arithmetic mean of the lifetimes of the donor-only samples measured in the same experiment: FRETeff = 1 − ($\tau_{FRET}/\tau_{donor}$). All measurements were performed in three independent experiments (n = 8).

### Yeast two-hybrid (Y2H) assays

Full-length CDS of MDF, LSM8, LSM2, and STA1 were amplified and cloned into pDONR221 plasmids. The gateway compatible versions of the GAL4 DNA-binding domain vector pGBT-9 (Bleckmann et al, 2010) and the activation domain vector pGAD424 (Clontech, www.takarabio.com) were used as destination vectors. *S. cerevisiae* strain AH109 was transformed as described in Gietz et al (1997). Positive transformants were selected on yeast minimal medium (SD medium: 0.66% yeast nitrogen base without amino 244 acids, 0.066% amino acid mix, 2% glucose), lacking leucine and tryptophan (SD-LW). Single positive colonies were cultured ON in liquid SD-LW at 30°C and continuous shaking. The day after a pre-culture with OD600 = 0.3 was inoculated using the ON culture as the starter material. After 3 h under continuous shaking at 30°C, optical density was adjusted to OD600 = 4, and a dilution series from 10-1 to 10-3 was made. Spotting of the pre-culture and dilutions was carried out on selection plates containing either SD-LW (growth control) or SD-LWH (interaction test). Three colonies were tested for each of the interactions.

### Confocal laser scanning microscopy

Detection of YFP, GFP, mCherry, and PI was carried out by using the Leica TCS SP8 Confocal Platform (Leica Microsystems). Excitation wavelength of 488 nm was used for detection of GFP and YFP, whereas a laser light of 561 was used for excitation of mCherry and PI. The detection windows ranged from 520–540 (YFP), 496–514 (GFP), 590–630 (mCherry), and 600–636 (PI). For BiFC analyses, images were collected using a TCS SP5 confocal laser scanning microscope (Leica Microsystems). The excitation line for imaging YFP fusions was 514 nm.

### RNA extraction and RT-qPCR analyses

Total RNA was extracted from complete seedlings of the different ages using the innuPREP Plant RNA kit (Analytik Jena Bio solutions). cDNA synthesis was carried out by using the QuantiTect Reverse Transcription Kit (QIAGEN). Rotor-Gene SYBR Green (QIAGEN) and the Rotor-gene-Q cycler (QIAGEN) were used for performing the quantitative PCRs. Generated data were quality-controlled and normalized to the reference genes *FASS* (*AT5G18580*) and *SAND* (*AT2G28390*) using the qbasePLUS software (Hellemans et al, 2007). For validation of intron retention events visualized on the Integrative Genomics Viewer software, two primers pairs were designed to

specifically detect intron and exon transcripts. Exon primers annealed to the exon junction, whereas intron-specific primers annealed inside the retained intron. Intron retention was quantified by dividing intron expression by exon-specific expression. All primers are listed on Table S16.

### RNA sequencing

Total RNA was extracted from three different biological replicates containing 12-d-old seedlings of WT and *mdf-1* using the innuPREP Plant RNA Kit (Analytik Jena Bio solutions). RNA quality assessment, library preparation, sequencing, and bioinformatics analyses were performed by Novogene Co. Integrity and quantitation of the extracted RNA were measured using the RNA Nano 6000 Assay Kit of the Bioanalyzer 2100 system (Agilent Technologies). With 1 μg RNA per sample as input material, sequencing libraries were subsequently generated using the NEBNext UltraTM RNA Library Prep Kit for Illumina (NEB) following the manufacturer's recommendations. Index codes were also added to attribute sequences to each sample. Clustering of the index-coded samples was carried out on a cBot Cluster Generation System using PE Cluster Kit cBot-HS (Illumina) following manufacturer's instructions. Afterward, library preparations were sequenced on an Illumina NovaSeq 6000 platform and ~50 million 150 paired end reads per sample were generated. To ensure the high quality of the samples, only clean data were used for subsequent analyses. This was achieved after the removal of reads containing adapter and poly-N sequences and reads with low quality from the raw data. Mapping against the *Arabidopsis* TAIR10 reference genome was performed using the HISAT2 software. Reads per kilobase of exon model per million mapped reads (RPKM) of each gene was calculated based on the length of the gene and reads count mapped to this gene. Differential expression analyses between *mdf-1* and WT were carried out using the DESeq2 R package with the Benjamini and Hochberg's approach for adjusting *P*-values according to the false discovery rate. Genes with adjusted *P*-value < 0.05 were considered as differentially expressed. Gene Ontology (GO) enrichment analyses was performed using the GO term finder software from Princeton University (Boyle et al, 2004) and the degree of significance of selected relevant terms ($-\log_{10}$-*P*adj) was graphically represented together with the number of genes (n) found in the different categories. Alternative splicing analysis was performed with the rMATS software. Events with adjusted *P*-value < 0.05 were considered as alternatively spliced. Venn diagrams were generated with the online software Venny 2.1 generated by Juan Carlos Oliveros (BioinfoGP, CNB-CSIC). Statistical significance of the overlaps represented was calculated based on a normal approximation of a hypergeometric probability formula implemented at the web tool http://nemates.org/MA/progs/overlap_stats.html. *Arabidopsis* reference number of protein-coding genes was set to 27474.

## Data Availability

The genome-wide RNAseq data presented in this publication are available on the Gene Expression Omnibus database GSE197898 (https://www.ncbi.nlm.nih.gov/geo/).

## Supplementary Information

## Acknowledgements

We thank Judith Mehrmann, Sotoodeh Seyedehyasaman, Wiebke Hellmeyer, Anastasiia Goshina, and Tim Lienemann for technical assistance and experimental support and to Paul Larsen (University of California, Riverside) for providing the seeds for the *sog1-7* line. This work was funded by the Deutsche Forschungsgemeinschaft (DFG) grant to M Weingartner (WE4506/6-1). C de Luxán-Hernández was recipient of a PhD fellowship from the University of Hamburg. In addition, funding was received from the Spanish Ministry of Science and Innovation to J Salinas (PID2019-106987RB-I00/AEI/10.13039/5011033), from the Spanish Ministry of Science and Innovation to E Tranque (FPI contract), from the Czech Academy of Science, Deutsche Akademische Austauschdienst, and Bundesministeriums für Bildung und Forschung (BMBF) (CAS Mobility DAAD-19-04 and CAS Mobility DAAD-21-07) to M Weingartner and P Binarova.

### Author Contributions

C de Luxán-Hernández: formal analysis, investigation, visualization, and writing—review and editing.
J Lohmann: formal analysis, investigation, and visualization.
E Tranque: formal analysis, investigation, and visualization.
J Chumova: formal analysis, investigation, and visualization.
P Binarova: conceptualization, formal analysis, supervision, funding acquisition, visualization, project administration, and writing—review and editing.
J Salinas: formal analysis, supervision, funding acquisition, visualization, project administration, and writing—review and editing.
M Weingartner: conceptualization, formal analysis, supervision, funding acquisition, writing—original draft, and project administration.

### Conflict of Interest Statement

The authors declare that they have no conflict of interest.

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
