## [Reviewer comments · Life Science Alliance]

Life Science Alliance

MDF is a conserved splicing factor and modulates cell division and stress response in Arabidopsis

Cloe de Luxán-Hernández, Julia Lohmann, Eduardo Tranque, Jana Chumova, Pavla Binarova, Julio Salinas, and Magdalena Weingartner

DOI: <https://doi.org/10.26508/lsa.202201507>

Corresponding author(s): Magdalena Weingartner, Universität Hamburg

Review Timeline:

Submission Date:	2022-04-28
Editorial Decision:	2022-06-01
Revision Received:	2022-09-06
Editorial Decision:	2022-09-23
Revision Received:	2022-09-27
Accepted:	2022-09-27

Transaction Report:

June 1, 2022

Re: Life Science Alliance manuscript #LSA-2022-01507-T

Magdalena Weingartner
University of Hamburg
Ohnhorststrasse 18
Hamburg, Germany 22609

Dear Dr. Weingartner,

Thank you for submitting your manuscript entitled "MDF is a conserved splicing factor and modulates cell division and stress response in Arabidopsis" to Life Science Alliance. The manuscript was assessed by expert reviewers, whose comments are appended to this letter. We invite you to submit a revised manuscript addressing the Reviewer comments.

Thank you for this interesting contribution to Life Science Alliance. We are looking forward to receiving your revised manuscript.

Sincerely,

B. MANUSCRIPT ORGANIZATION AND FORMATTING:

Reviewer #1 (Comments to the Authors (Required)):

In this manuscript, the authors reported the molecular function of a splicing factor, MDF, in Arabidopsis. MDF associates with spliceosome protein LSM8. Consistent with its annotation, loss of MDF leads to disrupted splicing of many genes, and various morphological and physiological phenotypes accompany this. The authors further proposed that MDF coordinates stress response and cell division. Overall, I believe the majority of the work is a presentation of high-quality and convincing data. In the meantime, some conclusion/hypothesis like MDF functions in coordinating stress response and cell division is quite indirect and need more evidence. As listed below, a few issues need to be addressed before publication.

1. Concerning Table 1 and Figure 2. The author described the association between MDF and LSM8. Logic-wise, it is not very clear to this reviewer why the author tested the interaction only between MDF and LSM8. Why weren't other components of the U4/U6.U5 tri-snRNP complex tested? In addition, the author finds that MDF does not directly interact with LSM8. How about the situation for SART1 and LSM8 in humans?
2. In association with point1, when testing the interaction between MDF and LSM8, the choice to do the LSM8-GFP IP-MS as a starting point is unfortunate. I recommend the author show MDF IP-MS results. This would provide a more unbiased picture of the interactome of MDF. The focus of this paper is MDF while not LSM8. This is important, especially since the author found that MDF only indirectly interacts with LSM8.
3. "Its accumulation in nuclear speckles further indicated that it was part of the plant splicing complex" This sentence is misleading as currently, there is no evidence to show the identity of speckles. Under a context of liquid-liquid phase separation, any protein with multivalent interactions can have this property.
4. Figure 3C, the splicing defective genes in *mdf-1* overlap not so well with that in *lsm8*, *rdm16*, or *brr2a*. Does that suggest MDF affects splicing largely independent of these components of spliceosome? This needs to be discussed.
5. Figure 3D, the value on the Y-axis is not clear.
6. The author found 7516 and 7872 genes up and down regulated, respectively. The cut-off here is quite loose for this reviewer. The difference at this number could be overestimated and misleading. How many of those changes are more than two fold compared with Col-0? Please revise the text and associated analysis accordingly.
7. Figure 4D. "Additionally, approximately 64% of the genes differentially intron retained in *mdf-1* background...." First of all, based on figure 4D, the overlap is much smaller than 64%. Second, the overlap with the upregulated genes is also significant or similar to that of downregulated genes. This part needs to be elaborated further.
8. Fig 5a, a statistical test needs to be performed to show whether the overlap between SOG1 targets and MDF1 targets is significant.
9. Sentence "...thus most likely contributed to its growth arrest phenotype" has no direct evidence to support it, so it should be moved into the discussion.
10. Related to Figure 7C, the author analyzed the MDF22A mutant transgenic line; I wonder how many lines the author studied and why this particular line was selected. Results from more than one line would be welcome.
11. Related to point 10, Figure 7F, the author analyzed the rescue of the splicing defects of *mdf* in MDF22A line. A sequencing analysis instead of just the qPCR would be welcome here.
12. Figure 8, the author studied how SR45 and MDF might influence the DDR pathway, the figure is OK, but the test towards the end of the main text is basically discussion-like. There is no evidence that the DDR defective phenotype in *sr45* or *mdf* is directly or indirectly linked with these proteins.
13. In the abstract, "correct splicing of numerous transcripts". Please be precise with the number.

Reviewer #2 (Comments to the Authors (Required)):

This manuscript describes studies of the Arabidopsis MDF gene that is a homolog of the human splicing factor SART1, a component of the U4/U6/U5 tri-snRNP. Previous studies have reported a role of MDF in primary root development and identified S22 phosphorylation of MDF. This study confirmed the defects in root elongation in two *mdf* mutants and pinpointed the defect to a cell cycle arrest at the G2/M transition in the root meristem. It was shown that MDF associates with LSM8 by BiFC but the association is unlikely to be direct. RNA-seq identified more than 2000 genes with splicing defects in *mdf-1*, with intron retention being the predominant defect. Meanwhile, thousands of up- and down-regulated genes were found in *mdf* mutants. About 50% of the genes with intron retention are also either up- or down-regulated in the *mdf-1* mutant. The RNA-seq analysis also

revealed the constitutive expression of genes involved in DNA damage repair (DDR), a well-studied pathway that leads to cell cycle arrest and cell death, phenotypes found in *mdf* mutants. In particular, two genes in the DDR pathway, ANAC044 and ANAC085, are increased in expression in *mdf* mutants. While SOG1 is known to activate ANAC044 and ANAC085 in DDR, a *sog1* mutation not only did not suppress the root phenotype of *mdf-1* but even enhanced the phenotype, suggesting that MDF and SOG1 likely act in parallel. By analyzing the transgene containing a S22A mutation in MDF, they authors showed that S22 phosphorylation is required in some of the molecular functions of MDF. The authors also tested the response of *mdf* mutants and two other splicing factor mutants (*lsm8* and *sr45*) to the double-stranded breaks inducing drug zeocin and found that these splicing factor mutants were hypersensitive to DNA damage.

Overall, the results reported in this study are solid and contribute to the understanding of the molecular and, to some extent, the developmental functions of MDF. The findings from the study, however, appear to be isolated and lack integration. For example, the defects of the *mdf* mutants in splicing and in gene expression are clear, but it is unknown whether the splicing defects lead to the expression defects. Similarly, it is also unknown whether the constitutive expression of DDR genes leads to the cell cycle arrest or cell death phenotypes. The authors tried to test whether the *sog1* mutation suppresses the *mdf* phenotype in cell death, but did not see suppression but rather saw enhancement. As such, manuscript appeared to consist of a number of unconnected pieces of information.

Major points

1. It is surprising that the authors did not examine the expression of ANAC044 and ANAC085 in the *mdf-1 sog1-7* double mutant. This would reveal whether the up-regulation of ANAC044 and ANAC085 expression in the *mdf-1* mutant requires SOG1.
2. Given that *sog1-7* did not suppress the cell death phenotype of *mdf-1* and that the expression of ANAC044 and ANAC085 is increased in *mdf-1*, the authors should have generated the *mdf-1 anac044 anac085* triple mutant to determine whether ANAC044 and ANAC085 are responsible for the phenotypes of *mdf-1*.
3. ANAC044 and ANAC085 are supposed to promote the expression of Rep-MYB. Is Rep-MYB expression affected in *mdf* mutant?
4. In Figure 1B, the authors determined the cell death in the cell division zone of the RAM using PI staining. It would be best to complement this assay staining for viability using fluorescein diacetate (FDA) and for nuclei using DAPI.
5. For Figure 1E, it is better to add representative images of microtubule immunostaining and of preprophase bands to show the overall difference between *Col* and *mdf*.
6. The last section of the manuscript describes hypersensitivity of splicing-related mutants, including *mdf* mutants, to zeocin that induces DNA double-stranded breaks. The hypersensitivity of *mdf* mutants is counter intuitive and the authors did not try to rationalize it. As the *mdf* mutants show constitutive DDR response, one would expect the mutants to be resistant to neocin, which induces DDR response. The observation that *mdf* mutants show reduced DDR response upon neocin treatment is surprising. What is the overlap of DDR genes activated by neocin in WT vs. those constitutively expressed in *mdf* mutants?

Minor points

As the results of the rescue experiment with the wild-type and S22A transgene were not clear-cut (even the wild-type version did not fully rescue the mutant and the S22A mutant version rescued some but not all of the molecular defects), the authors should tone down the conclusion that S22 phosphorylation is important.

Better description of the phenotypic characterization of cell division or cell death in roots is needed. For example, was confocal imaging performed with Z-stacks? How were the images analyzed?

Reviewer #3 (Comments to the Authors (Required)):

In this work, Luxan-Hernandez and coworkers explore the function of MDF in *Arabidopsis thaliana*. MDF homology with the human protein START1 first served to predict MDF function in splicing regulation. However, this remained only an assumption. In this work, the authors combined cell biology, molecular and genetic approaches to demonstrate the role of MDF in alternative splicing. This manuscript is very well written (a very few typos are indicated in the end as minor comments) and it's easy to follow. As an interested reader in alternative splicing, I'd be happy to see this piece of work published soon.

I still have a couple of suggestions to further improve the manuscript prior to publication.

1. By complementing the *mdf* mutant with WT vs MDF22A mutant, the authors conclude that phosphorylation is relevant for the protein function. What does it change in the protein? Comparing with the role of this post-translational modification of splicing factors in animals (e.g. SRSF1), does it affect MDF stability (western blot needed, perhaps)? Nuclear-cytoplasmic distribution? I'd like to see protein (wt and mutant) sub cellular localization in control vs zeocin treatment.
2. Authors showed the interaction between MDF and the splicing factor LSM8. Similarly to point 1, does DNA damage induction affect the interaction?
3. Alternative splicing was assessed on RNA-seq data by using the rMATS software. In our hands (see Rigo et al 2020, EMBO Rep), different softwares may deliver different results. Have you tried other tools, including RNAProf or IsoSwitch? The type of data delivered is different and more information may be retrieved (i.e. protein domains included/excluded, NMD fate due to novel

stop codons within the retained intron, etc). I suggest trying, if the authors agree.

Minor comments:

As a fan of lncRNAs, I'd have liked to find a deeper discussion about the interplay between SFs, noncoding transcripts, phosphorylation and even nuclear sublocalization. We wrote a review article a couple of years ago which may serve as an introductory approach into this field (Romero-Barríos et al 2018, NAR). Of course, this is only a suggestion, please feel free to ignore it!

There are a few typos along the text:

12-day-old is once written 12-dayS-old, and it should be singular.

The phrase "Compared to WT, mdf-1, lsm8-1 and sr45-1 were significantly stronger affected by presenting complete growth arrest already four days after transfer to zeocin supplemented media (Figure 8B)." is weird. "stronger" doesn't fit here, "more" can be used instead, as in a phrase later in the manuscript.

The phrase "...whose loss of function was associated with several developmental defects such late flowering, reduced branching" it should say "such as".

Congratulations for this article, I enjoyed reading it.

Federico Ariel

Reviewer #1 (Comments to the Authors (Required)):

General Statement:

Overall, I believe the majority of the work is a presentation of high-quality and convincing data. In the meantime, some conclusion/hypothesis like MDF functions in coordinating stress response and cell division is quite indirect and need more evidence. As listed below, a few issues need to be addressed before publication.

1. Concerning Table 1 and Figure 2.

The author described the association between MDF and LSM8. Logic-wise, it is not very clear to this reviewer why the author tested the interaction only between MDF and LSM8. Why weren't other components of the U4/U6.U5 tri-snRNP complex tested?

In addition, the author finds that MDF does not directly interact with LSM8. How about the situation for SART1 and LSM8 in humans?

To our knowledge physical interaction between LSM8 and SART1 (human) or SNU66 (yeast) has not been tested so far. We found that MDF co-immunoprecipitated with LSM8, we could confirm that they act in the same complex, however we found that MDF and LSM8 do not physically interact.

It was shown in Y2H experiments that human SART1 physically interacts with hBRR2A and hPERP6 (U5-associated proteins) and also PRP3 (U4/U6-associated protein). In yeast, ySNU66 was shown to physically interact with yPRP8 and yBRR2 (U5-associated proteins). We have performed additional Y2H hybrid experiments to find out if MDF physically interacts -like SART1 - with the plant homolog of PRP6, which is in Arabidopsis named STA1 (STABILIZED 1). Indeed, we found that MDF interacts with STA1. These results are now shown in Figure 2D and S2B.

2. In association with point1, when testing the interaction between MDF and LSM8, the choice to do the LSM8-GFP IP-MS as a starting point is unfortunate. I recommend the author show MDF IP-MS results. This would provide a more unbiased picture of the interactome of MDF. The focus of this paper is MDF while not LSM8. This is important, especially since the author found that MDF only indirectly interacts with LSM8.

The goal of this work was to show that MDF, like its human counterpart SART1 is a spliceosome protein. Therefore, the finding that MDF is an interaction partner of LSM8 was interesting and we

could confirm that both proteins act in the same complex by performing BiFC experiments. We have now tested interaction of MDF with other components of the complex and found - using Y2H - that MDF interacts with STA1.

3. "Its accumulation in nuclear speckles further indicated that it was part of the plant splicing complex" This sentence is misleading as currently, there is no evidence to show the identity of speckles. Under a context of liquid-liquid phase separation, any protein with multivalent interactions can have this property.

We fully agree with the reviewer that our text was misleading since we had not confirmed that the observed nuclear foci contained known splicing factors. Therefore, we have performed additional experiments by co-expressing the MDF-GFP fusion protein with SR45-mCherry, a well-established marker for nuclear speckles in plants. Our results shown in Figure 2E confirmed that both proteins co-localised in nuclear condensates.

4. Figure 3C, the splicing defective genes in *mdf-1* overlap not so well with that in *lsm8*, *rdm16*, or *brr2a*. Does that suggest MDF affects splicing largely independent of these components of spliceosome? This needs to be discussed.

A paragraph in which this issue is being discussed in more detail has been added to the discussion.

5. Figure 3D, the value on the Y-axis is not clear.

Although the methodological approach followed during this study to obtain the intron retention values was previously briefly described in the material and methods section, we agree with the reviewer that more clarification was needed. Therefore, we have added in the legend to Figure 3D-G a text explaining how these values were calculated.

6. The author found 7516 and 7872 genes up and down regulated, respectively. The cut-off here is quite loose for this reviewer. The difference at this number could be overestimated and misleading. How many of those changes are more than two fold compared with Col-0? Please revise the text and associated analysis accordingly.

We agree with this reviewer that this was not explicitly explained. The 7516 up and 7872 down regulated genes are loci that show significantly altered transcript levels in *mdf-1* compared to WT without considering any threshold for fold change. We followed this approach to obtain a broader picture of the expression changes associated to the loss of MDF since only one timepoint (12 days) and whole mount samples were used for the RNA-seq experiments. Based on that, we tried to avoid introducing any bias by only considering differentiated genes highly expressed at that specific timepoint. It would be interesting in the future to perform a time series so that the transcriptional profile at earlier developmental stages and individual tissues can be analyzed in detail, and here we agree with the reviewer that a higher threshold would be required.

Nevertheless, as specified in the text, for the comparative analysis of transcriptional changes associated with genome stability (by comparing against SOG1 targets) a Log2Fold Threshold of above 2 was already set, to further support the importance of MDF in cell division, DNA repair and cell death in the context of DNA damage (Figure 5).

7. Figure 4D. "Additionally, approximately 64% of the genes differentially intron retained in *mdf-1* background...." First of all, based on figure 4D, the overlap is much smaller than 64%. Second, the overlap with the upregulated genes is also significant or similar to that of downregulated genes. This part needs to be elaborated further.

We have revised the table in Figure 4D to show in a more detailed manner the overlap between genes showing IR and differential expression, IR and up-regulation, IR and down-regulation or only IR as well as the statistical significance of the overlap. The text has also been changed accordingly.

8. Fig 5a, a statistical test needs to be performed to show whether the overlap between SOG1 targets and MDF1 targets is significant.

We have calculated the statistical significance of the overlap between DEG genes in *mdf-1* and SOG1 target genes and Rep-MYB target genes respectively by considering as total amount of genes the Arabidopsis reference number of 27474 protein coding genes. These results have been added in the text and the statistical approach followed to calculate the statistical significance is described in the Figure legend (Figure 5).

9. Sentence "...thus most likely contributed to its growth arrest phenotype" has no direct evidence to support it, so it should be moved into the discussion.

We have changed the text accordingly.

10. Related to Figure 7C, the author analyzed the MDF22A mutant transgenic line; I wonder how many lines the author studied and why this particular line was selected. Results from more than one line would be welcome.

We have generated 6 independent lines expressing the *p35SMDF* construct and 2 independent lines expressing the *p35SMDFS22A* construct in the *mdf-1* mutant background. The results shown in Figure 7 were generated using a mixed population of progeny from each of the transgenic lines for each construct generated. To further confirm these results, we performed additional experiments using progeny from two independent lines for each construct and obtained the same results. These additional results are shown now in Supplemental Figure S4.

11. Related to point 10, Figure 7F, the author analyzed the rescue of the splicing defects of *mdf* in MDF22A line. A sequencing analysis instead of just the qPCR would be welcome here.

We fully agree with this reviewer that a genome wide sequencing analysis of both transgenic lines would provide very interesting new datasets that would further substantiate our conclusions. However, at the moment we do not have these data. We believe that our phenotypic data as well as our RT-qPCR analyses provide already important insight into how the phosphorylation pattern of MDF might influence its activity.

12. Figure 8, the author studied how SR45 and MDF might influence the DDR pathway, the figure is OK, but the test towards the end of the main text is basically discussion-like. There is no evidence that the DDR defective phenotype in *sr45* or *mdf* is directly or indirectly linked with these proteins.

The text of this part of the results section has been changed and the discussion-like passage was removed from the Results.

13. In the abstract, "correct splicing of numerous transcripts". Please be precise with the number.

We have changed the text of the abstract accordingly.

Reviewer #2 (Comments to the Authors (Required)):

This manuscript describes studies of the Arabidopsis MDF gene that is a homolog of the human splicing factor SART1, a component of the U4/U6/U5 tri-snRNP. Previous studies have reported a role of MDF in primary root development and identified S22 phosphorylation of MDF. This study confirmed the defects in root elongation in two *mdf* mutants and pinpointed the defect to a cell cycle arrest at the G2/M transition in the root meristem. It was shown that MDF associates with LSM8 by BiFC but the association is unlikely to be direct. RNA-seq identified more than 2000 genes with splicing defects in *mdf-1*, with intron retention being the predominant defect. Meanwhile, thousands of up- and down-regulated genes were found in *mdf* mutants. About 50% of the genes with intron retention are also either up- or down-regulated in the *mdf-1* mutant. The RNA-seq analysis also revealed the constitutive expression of genes involved in DNA damage repair (DDR), a well-studied pathway that leads to cell cycle arrest and cell death, phenotypes found in *mdf* mutants. In particular, two genes in the DDR pathway, ANAC044 and ANAC085, are increased in expression in *mdf* mutants. While SOG1 is known to activate ANAC044 and ANAC085 in DDR, a *sog1* mutation not only did not suppress the root phenotype of *mdf-1* but even enhanced the phenotype, suggesting that MDF and SOG1 likely act in parallel. By analyzing the transgene containing a S22A mutation in MDF, they authors showed that S22 phosphorylation is required in some of the molecular functions of MDF. The authors also tested the response of *mdf* mutants and two other splicing factor mutants (*lsm8* and *sr45*) to the double-stranded breaks inducing drug zeocin and found that these splicing factor mutants were hypersensitive to DNA damage.

Overall, the results reported in this study are solid and contribute to the understanding of the molecular and, to some extent, the developmental functions of MDF. The findings from the study, however, appear to be isolated and lack integration. For example, the defects of the *mdf* mutants in splicing and in gene expression are clear, but it is unknown whether the splicing defects lead to the expression defects. Similarly, it is also unknown whether the constitutive expression of DDR genes leads to the cell cycle arrest or cell death phenotypes. The authors tried to test whether the *sog1* mutation suppresses the *mdf* phenotype in cell death, but did not see suppression but rather saw enhancement. As such, manuscript appeared to consist of a number of unconnected pieces of information.

Major points

1. It is surprising that the authors did not examine the expression of ANAC044 and ANAC085 in the *mdf-1 sog1-7* double mutant. This would reveal whether the up-regulation of ANAC044 and ANAC085 expression in the *mdf-1* mutant requires SOG1.

This is indeed a very interesting suggestion. We have now performed additional RT-qPCR analyses and tested the expression of ANAC044 and ANAC085 in the single mutant and the *mdf sog1-7* double mutant. We found that the overexpression of both transcription factors is maintained in the *mdf-1sog1-7* double mutants and thus occurs independent of SOG1. These data are presented in Figure 6C of the revised manuscript.

2. Given that *sog1-7* did not suppress the cell death phenotype of *mdf-1* and that the expression of ANAC044 and ANAC085 is increased in *mdf-1*, the authors should have generated the *mdf-1 anac044*

anac085 triple mutant to determine whether ANAC044 and ANAC085 are responsible for the phenotypes of *mdf-1*.

We agree with this reviewer that this would be a very interesting analysis. However, the aim of this study was to show that the *mdf-1* mutation is associated with mis-splicing and mis-expression of numerous transcripts including those for genes involved in cell cycle control. We used the ANAC transcription factors as an example to prove that important cell cycle regulators are indeed mis-expressed in an MDF-dependent manner. Given the fact that more than 270 cell cycle associated genes are mis-expressed in the *mdf-1* mutant, we do not expect that loss of function of these two transcription factors alone would rescue the *mdf-1* mutant phenotype. Additionally, previous publications showed how *ANAC044* overexpression by itself could not induce cell cycle arrest (Takahashi *et al.*, 2019)

3. ANAC044 and ANAC085 are supposed to promote the expression of Rep-MYB. Is Rep-MYB expression affected in *mdf* mutant?

In our RNA-seq data the REP-MYB genes do not appear to be transcriptionally changed. It was shown previously (Takahashi *et al.*, 2019) that the REP-MYB transcription factors are upon DNA damage mainly regulated at the protein level and it was therefore suggested that the NAC-type transcription factors might indirectly regulate their protein stability. Additionally, the REP-MYBs were not found among the genes transcriptionally changed after zeocin treatment in WT plants (Yoshiyama *et al.*, 2020).

Interestingly we found that at least one of the MYB-type transcription factors was mis-spliced in the *mdf-1* mutant. Additional experiments will be required to find out if the abundance of REP-MYB transcription factors are also regulated by alternative splicing.

4. In Figure 1B, the authors determined the cell death in the cell division zone of the RAM using PI staining. It would be best to complement this assay staining for viability using fluorescein diacetate (FDA) and for nuclei using DAPI.

We have performed FDA staining of root tips of WT, *mdf-1* and the *mdf-1* complementation line which confirm that cell death occurred only in the meristematic zone of *mdf-1*. Representative images are shown in Figure S1A.

5. For Figure 1E, it is better to add representative images of microtubule immunostaining and of preprophase bands to show the overall difference between Col and *mdf*.

We agree with the reviewer that images of immunostaining for microtubules in root tip cells showing reduced number of mitotic microtubular arrays and accumulation of PPBs in *mdf-1* were missing. The images are included in the revised version of the manuscript as Figure S1B and we believe that they demonstrate better overall difference between WT and *mdf-1* seedlings.

6. The last section of the manuscript describes hypersensitivity of splicing-related mutants, including *mdf* mutants, to zeocin that induces DNA double-stranded breaks. The hypersensitivity of *mdf* mutants is counter intuitive and the authors did not try to rationalize it. As the *mdf* mutants show constitutive DDR response, one would expect the mutants to be resistant to neocin, which induces DDR response. The observation that *mdf* mutants show reduced DDR response upon neocin treatment is surprising. What is the overlap of DDR genes activated by neocin in WT vs. those constitutively expressed in *mdf* mutants?

We have analyzed the overlap between zeocin induced transcriptional changes in WT (Yoshiyama *et al.*, 2020) and transcripts that are significantly differentially expressed in the *mdf-1* mutant under control conditions. We found that out of 1912 genes reported to be transcriptionally regulated after zeocin incubation, there is a significant overlap of 1353 genes also undergoing expression changes in the absence of MDF. These data have been added as Figure 5A to the revised manuscript.

We agree with this reviewer that the issues about how constitutive expression of DDR genes under control conditions and the impaired induction of DNA repair genes upon zeocin treatment as well as the zeocin-hypersensitive phenotype might be explained were not sufficiently discussed in the first version of this manuscript. Therefore, we have added an additional paragraph about this topic to the discussion.

Minor points

As the results of the rescue experiment with the wild-type and S22A transgene were not clear-cut (even the wild-type version did not fully rescue the mutant and the S22A mutant version rescued some but not all of the molecular defects), the authors should tone down the conclusion that S22 phosphorylation is important.

We have adjusted the text in the results section accordingly.

Better description of the phenotypic characterization of cell division or cell death in roots is needed. For example, was confocal imaging performed with Z-stacks? How were the images analyzed?

We agree with the reviewer that more clarification was needed. For quantification analyses of number of dividing cells and cell death area in PI-stained root tips, single layer images were made by focusing on the same focal plane in which the QC is visible. Analysis of dividing cells was performed by counting the cells in the cortical cell layer. The details about how these analyses were performed have also been added to the materials and methods section of the revised manuscript.

Reviewer #3 (Comments to the Authors (Required)):

In this work, Luxan-Hernandez and coworkers explore the function of MDF in *Arabidopsis thaliana*. MDF homology with the human protein START1 first served to predict MDF function in splicing regulation. However, this remained only an assumption. In this work, the authors combined cell biology, molecular and genetic approaches to demonstrate the role of MDF in alternative splicing. This manuscript is very well written (a very few typos are indicated in the end as minor comments) and it's easy to follow. As an interested reader in alternative splicing, I'd be happy to see this piece of work published soon.

I still have a couple of suggestions to further improve the manuscript prior to publication.

1. By complementing the *mdf* mutant with WT vs MDF22A mutant, the authors conclude that phosphorylation is relevant for the protein function. What does it change in the protein? Comparing with the role of this post-translational modification of splicing factors in animals (e.g. SRSF1), does it affect MDF stability (western blot needed, perhaps)? Nuclear-cytoplasmic distribution?

We have analyzed the YFP-associated fluorescence in root tips of the *mdf-1::p35SMDF* and *mdf-1::p35SMDFS22A* in both of which the cDNA of MDF is C-terminally fused to the YFP. However, we did not find any changes in the sub-nuclear or sub-cellular distributions. Since representative confocal images, in which the YFP-fluorescence of both constructs are already included in the manuscript (Figure 7B) we did not add any additional data.

In addition, we analyzed whether MDF protein stability might be affected by the phosphorylation status by comparing YFP-derived fluorescence between 3 dag *mdf-1::p35SMDF* and *mdf-1::P35SMDFS22A* lines at control conditions and after treatment with the proteasome inhibitor Mg132 for 3 hours at a concentration of 100µM. No significant change was observed. These data are shown in an attached document (Figure A1 for Reviewers), but they have not been included in the manuscript.

I'd like to see protein (wt and mutant) sub cellular localization in control vs zeocin treatment.

This is a very nice and interesting suggestion. We have analyzed the YFP fluorescence in root tips of both lines under control conditions and after treatment with zeocin. However, we did not find any changes in the sub-nuclear or sub-cellular distributions. Representative confocal images are shown in Figure S5 of the revised manuscript.

2. Authors showed the interaction between MDF and the splicing factor LSM8. Similarly, to point 1, does DNA damage induction affect the interaction?

Although this would be interesting experiment to do, it seems at the moment technically difficult. As we found that MDF and LSM8 are acting in the same complex but do not physically interact, following how the indirect interaction is affected during DNA damage will be not an easy task.

3. Alternative splicing was assessed on RNA-seq data by using the rMATS software. In our hands (see Rigo et al 2020, EMBO Rep), different softwares may deliver different results. Have you tried other tools, including RNAProf or IsoSwitch? The type of data delivered is different and more information may be retrieved (i.e. protein domains included/excluded, NMD fate due to novel stop codons within the retained intron, etc). I suggest trying, if the authors agree.

Alternative splicing events were calculated with the rMATS software by an external company since it is their established pipeline for splicing analyses. Since our bioinformatic expertise is not the strongest one, we would rather keep with their data analysis. We thank reviewer for the important comments and for future projects concerning the splicing activity of MDF, we will perform the analyses suggested by the reviewer.

Minor comments:

As a fan of lncRNAs, I'd have liked to find a deeper discussion about the interplay between SFs, noncoding transcripts, phosphorylation and even nuclear sublocalization. We wrote a review article a couple of years ago which may serve as an introductory approach into this field (Romero-Barrios et al 2018, NAR). Of course, this is only a suggestion, please feel free to ignore it!

There are a few typos along the text:

12-day-old is once written 12-dayS-old, and it should be singular.

The phrase "Compared to WT, *mdf-1*, *lsm8-1* and *sr45-1* were significantly stronger affected by presenting complete growth arrest already four days after transfer to zeocin supplemented media (Figure 8B)." is weird. "stronger" doesn't fit here, "more" can be used instead, as in a phrase later in the manuscript.

The phrase "...whose loss of function was associated with several developmental defects such late flowering, reduced branching" it should say "such as".

Each of these typos have been corrected.

[Figure removed by LSA Editorial Staff per authors' request]

The phosphorylation status of MDF does not influence its protein stability. (A) Representative confocal pictures of PI-stained root tips cells of seedlings at 3 dag of *mdf-1::p35SMDF* and *mdf-1p35SMDFS22A* after incubation in 1% DMSO (Control-Upper panel) or in Mg132 100 μ M (Mg132-Lower panel) for 3 hours. YFP-derived fluorescence is shown in yellow. Scale bar: 50 μ M. (B) Corrected Total Fluorescence from the plants represented in (A). Fluorescence intensity was quantified by Image J for the whole RAM and corrected by the area and background of each individual transgenic line analyzed.

September 23, 2022

RE: Life Science Alliance Manuscript #LSA-2022-01507-TR

Dr. Magdalena Weingartner
Universität Hamburg
Ohnhorststrasse 18
Hamburg, Germany 22609
Germany

Dear Dr. Weingartner,

Thank you for submitting your revised manuscript entitled "MDF is a conserved splicing factor and modulates cell division and stress response in Arabidopsis". We would be happy to publish your paper in Life Science Alliance pending final revisions necessary to meet our formatting guidelines.

Along with points mentioned below, please tend to the following:
-the deposited RNA-seq data should now be made publicly available
-please consider uploading Figure 9 as a Graphical Abstract rather than as a figure.

A. FINAL FILES:

B. MANUSCRIPT ORGANIZATION AND FORMATTING:

****It is Life Science Alliance policy that if requested, original data images must be made available to the editors. Failure to provide original images upon request will result in unavoidable delays in publication. Please ensure that you have access to all original**

data images prior to final submission.**

The license to publish form must be signed before your manuscript can be sent to production. A link to the electronic license to publish form will be sent to the corresponding author only. Please take a moment to check your funder requirements.

Sincerely,

Reviewer #2 (Comments to the Authors (Required)):

The authors addressed my previous comments through additional experiments and discussions. I have no further comments.

September 27, 2022

RE: Life Science Alliance Manuscript #LSA-2022-01507-TRR

Dr. Magdalena Weingartner
Universität Hamburg
Ohnhorststrasse 18
Hamburg, Germany 22609
Germany

Dear Dr. Weingartner,

Thank you for submitting your Research Article entitled "MDF is a conserved splicing factor and modulates cell division and stress response in Arabidopsis". It is a pleasure to let you know that your manuscript is now accepted for publication in Life Science Alliance. Congratulations on this interesting work.

DISTRIBUTION OF MATERIALS:

Again, congratulations on a very nice paper. I hope you found the review process to be constructive and are pleased with how the manuscript was handled editorially. We look forward to future exciting submissions from your lab.

Sincerely,
